# Programmable liquid-core fibers: Reconfigurable local dispersion control for computationally optimized ultrafast supercontinuum generation

Johannes Hofmann [1], Ramona Scheibinger[1], Bennet Fischer [1], Mario Chemnitz [1,2] & Markus A. Schmidt [1,3] ✉

The field of computationally controlled light faces a strong demand for new platforms capable of providing adaptable light generation to meet the requirements of advanced photonic technologies. Here, we present the concept of computationally optimized nonlinear frequency conversion in programmable liquid-core fibers that enables real-time tunable and reconfigurable nonlinear power distribution through computationally optimized dispersion landscapes. The concept combines a temperature-sensitive mode in a liquid-core fiber, particle swarm optimization, fission of ultra-fast solitons, and a computer-controlled heating array to create a feedback loop for controlling output spectra via local temperature-induced dispersion modulation. Experiments and simulations show significant improvements in spectral power density over multiple predefined intervals simultaneously and broadband improved spectral flatness, highlighting the robustness and adaptability of the system. Beyond supercontinuum generation, the platform offers broad applicability to phenomena such as harmonic generation, soliton dynamics, spectral filtering, and multimode and hybrid fiber systems, opening up exciting opportunities for fundamental research and advanced photonic technologies.

The generation of light with tailored properties is vital in nearly all areas of modern photonics, with programmable reconfigurability, required to meet specific needs for applications such as spectroscopy[1,2], microscopy[3,4], metrology[5-7], communications[8-10], and quantum computing[11,12], representing a key unmet challenge.

In this context, ultra-fast supercontinuum generation (SCG) serves as a highly efficient method to produce tailored broadband light, with optical fibers providing an excellent platform due to their extended light-matter interaction lengths, strong core confinement, and tunable dispersion landscape[13]. Of particular interest are the

fission of temporal solitons and the associated emission of excess energy via dispersive waves (DWs), which is highly dependent on the group velocity dispersion (GVD) of the underlying fiber. Since the relevant dynamics are spatially defined, nonlinear conversion can be controlled by locally adapting the fiber dispersion. In this way, the fiber output power can be selectively enhanced in various spectral regions.

The ability to selectively manipulate local dispersion properties provides a unique feature by allowing multiple adjustments of nonlinear frequency conversion processes during pulse propagation, thereby extending spectral tuning possibilities far beyond those

[1]Leibniz Institute of Photonic Technology, Albert-Einstein-Str. 9, Jena, Germany. [2]Friedrich Schiller University Jena, Institute of Applied Optics and Biophysics, Philosophenweg 7, Jena, Germany. [3]Friedrich Schiller University Jena, Otto Schott Institute of Materials Research, Fraunhoferstr. 6, Jena, Germany. ✉e-mail: markus-alexander.schmidt@uni-jena.de

**Fig. 1 | Concept of computationally optimized nonlinear frequency conversion in programmable liquid-core fibers (pLCFs).** A temperature-sensitive mode propagating in a pLCF enables soliton-based supercontinuum generation (SCG), which, in combination with a computer-controlled heating array and particle swarm optimization, forms a feedback loop for precise spectral control. This approach allows algorithmic optimization of the supercontinuum output through local temperature-induced dispersion modulation along the fiber, enabling tunable and reconfigurable power transfers to selected wavelength intervals.

achieved with global dispersion control. Notable examples include Fiber-Bragg-Gratings[14,15], longitudinally varied fiber segments[16–20], and the periodic inclusion of waveguide resonances[21,22]. However, although successfully applied, the dispersion landscapes achieved through these techniques are static, resulting in non-reconfigurable nonlinear frequency conversion.

Reconfigurable methods for modifying fiber properties include acousto-optic tuning using external acoustic waves to induce microbending and gratings[23–25], electro-optic tuning by external electric fields[26–28], gas pressure tuning[29–31], and mechanical stress[32–34]. While these techniques allow for dispersion modulation, they are fundamentally limited in terms of spatial precision and the range of tunability that can be achieved.

In this context, liquid core fibers (LCFs) represent a novel nonlinear photonic platform for local and reconfigurable dispersion control[35]. These fibers encapsulate high refractive index liquids within a fiber-like structure[36] and have demonstrated exceptional effectiveness for broadband SCG. Key features include non-instantaneous nonlinear response, wide mid-IR transmission windows, and tunable dispersion achieved through composite liquids. Of particular interest is the high thermal coefficient of liquids such as carbon disulfide ($CS_2$)[37], comparable to that of semiconducting materials, which allows precise local dispersion control by temperature modulation. As a result, LCFs provide a unique and power-efficient platform for programmable control of soliton splitting and DW formation[35,38,39].

Optimization algorithms such as genetic algorithms (GAs) and particle swarm optimization (PSO) have recently been successfully used to control ultrafast SCG via input pulse shaping and mechanical deformations in multimode fibers[4,40–43]. It is important to note that these approaches rely on either fixed fiber dispersion or highly multimode operation, which may limit the amount of spectral modulation that can be achieved or lead to undesired complex output mode fields. These aspects motivate the investigation of alternative photonic platforms that offer greater flexibility in dispersion modulation for reconfigurable and programmable control of nonlinear processes.

In this work, we present the concept of computationally optimized nonlinear frequency conversion in programmable liquid-core fibers (pLCFs) for tunable and reconfigurable power transfers to multiple user-defined wavelength intervals through computationally optimized dispersion. Figure 1 illustrates this approach. The concept combines a temperature-sensitive spatial mode in a pLCF, particle

swarm optimization, soliton-based SCG and a computer-controlled heating array to form a feedback loop for precise spectral control of the SCG. This combination allows algorithmic optimization of the supercontinuum output, as demonstrated by both simulations and experiments. Hence, our study showcases the potential of computationally optimized dispersion profiles for advanced programmable light control in nonlinear single-mode waveguide systems.

## Results

Key to the concept of computationally optimized SCG in pLCF is the pronounced temperature sensitivity of the higher-order mode $TM_{01}$ in LCFs, which allows the nonlinear conversion processes to be controlled externally owing to the exceptional zero-dispersion shifts (cf. inset of Fig. 1). In contrast to previous studies[39], this work uses finely resolved temperature patterns along the fibers to locally modulate the dispersion and thus flexibly control the output spectra. When integrated with optimization algorithms, this implementation of a programmable fiber concept enables precise self-adaptation of the output spectral distribution.

### Soliton-based supercontinuum generation

The nonlinear frequency conversion scheme exploits soliton fission and the emission of multiple phase-matched DWs, a process that can be designed highly sensitive to the GVD of a mode of interest. Our study uses the $TM_{01}$, which exhibits a temperature-sensitive GVD with two zero-dispersion wavelengths (ZDWs) enclosing a region of anomalous dispersion (AD) at telecom wavelengths[44]. An ultra-short laser pulse injected into this region generates higher-order temporal solitons that decay to fundamental solitons while transferring excess energy to the emitted phase-matched DWs in both the near-IR (e.g. below 1400 nm) and the mid-IR (e.g., above 2000 nm).

To illustrate the pLCF concept, Fig. 2 shows the simulated spatio-spectral pulse evolution propagating in the $TM_{01}$ mode in a $CS_2$-filled LCF with 3.92 $\mu$m core diameter for two configurations of random dispersion landscapes (Fig. 2a, b) compared to the situation with constant dispersion at room temperature (Fig. 2c). These simulations are based on solving the generalized nonlinear Schrödinger equation, taking into account the varying modal properties along the fiber and the input pulse parameters (see Methods and Figure caption). Due to the arrangement of the heating elements used in the experiments (24 Peltier elements, approx. length 2.8 mm each), the GVD changes discretely, resulting in a stepwise variation of the ZDWs, indicated by

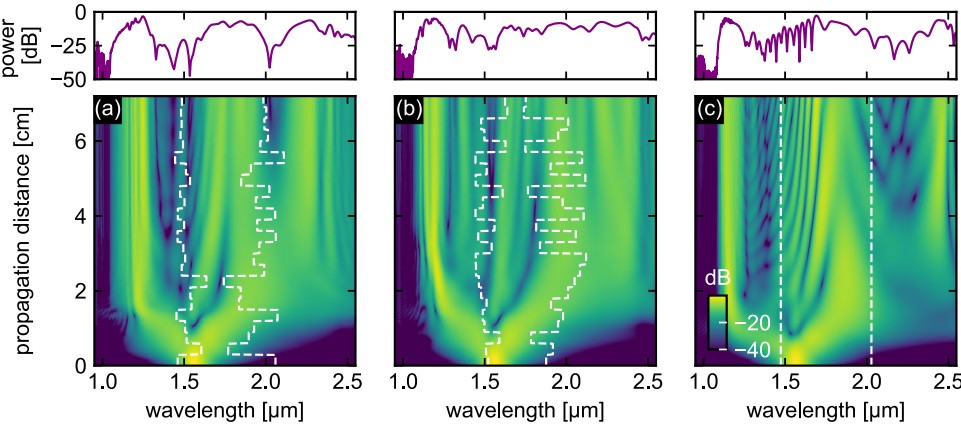

**Fig. 2 | Comparison of simulated spatio-spectral pulse evolutions of two temperature configurations of a CS2-filled LCF (core diameter: 3.92 μm, TM$_{01}$ mode). a, b** Pulse evolutions of random dispersion landscape. **c** Pulse evolution at room temperature. The white dashed lines show the behavior of the ZDWs surrounding an AD region. Note that the step wise change of ZDWs results from the experimental conditions involving Peltier elements of 2.8 mm length. The upper plots show the corresponding output spectra at a distance z = 7.2 cm. The simulations are based on an input pulse with $λ_p$ = 1560 nm, $τ$ = 30 fs and $P_{peak}$ = 10 kW. All simulations are normalized to the maximum intensity of the input pulse.

dashed white lines in Fig. 2. It should be noted that additional 2D finite-element simulations of the temperature distribution in a comparable silica glass block for various LCF geometries (see the Supplementary Note 1) reveal only minor deviations from a step-like longitudinal temperature profile when using the experimental fiber parameters, supporting the validity of the step-like temperature distribution assumption. Example nonlinear pulse propagation simulations revealed that the generated output spectra are largely insensitive to the specific shape of the temperature transition at the heating element-air interface, although future studies with full statistical analysis of different transition profiles are needed to comprehensively quantify this effect. Significantly different nonlinear dynamics (lower plots) and output spectra (upper plots) are observed in the three cases, highlighting the impact of dispersion variations on the output spectra and forming the scientific background for the concept. Note that the temperature distribution affects not only the first soliton fission but also the spectral distribution at longer propagation distances in a complex and non-intuitive way, justifying the need for computational optimization.

## Algorithmic approach

The computational optimization of spectral power addressed in this study is based on particle swarm optimization, a metaheuristic algorithm that explores an $N$-dimensional search space with $M$ particles. Optimization algorithms such as Genetic Algorithms (GAs) and Particle Swarm Optimization (PSO) have recently been used in reconfigurable nonlinear frequency translation to control SCG via pulse shaping[40–42]. Despite successful demonstrations, input pulse shaping, however, may offer changing coupling efficiency and flexibility in creating tailored pulse shapes and hence spectral control. While SCG is highly dependent on input pulse characteristics, the system's evolution dynamics enforces a specific energy flow prohibiting a wide range of nonlinear coupling further down the propagation. This highlights the need for a novel photonic platform that provides greater flexibility, and cost-effectiveness for reconfigurable and programmable control of nonlinear processes.

In our realization, each particle represents a potential solution corresponding to the number of Peltier elements. With fixed laser parameters, the swarm dynamically adapts on the basis of position and inertia, and optimizes spectral properties using tailored objective functions. PSO was chosen for its effectiveness in dealing with nonlinear dynamics, irregular solution landscapes, and multiple local

minima, providing robust performance without relying on initial parameter estimates or prior knowledge of the system.

Computational optimization is implemented through a feedback loop in which the PSO iteratively refines the objective function by comparing each solution to the target objective (schematically shown in Fig. 3). Note that the main difference between simulation and experiment is how the output spectra are obtained. In the simulation (green branch in Fig. 3), fiber dispersion and effective mode area are calculated, followed by nonlinear pulse propagation simulations based on the Generalized Nonlinear Schrödinger Equation. More details on the simulation framework and parameters can be found in the Methods section. In the experiment instead (yellow branch in Fig. 3), the fiber temperature is adjusted and the resulting output spectrum is measured by an automated data acquisition and control system, leading to a framework that operates independently of nonlinear pulse propagation simulations and relies exclusively on experimentally accessible parameters. Details on fiber preparation and optical setup can be found in the Methods section.

## Definition of objective functions

The choice of an appropriate objective function is crucial to achieving the desired results. In this study, the PSO aims to optimize either of the two set optimization objectives, i.e., (O1) maximizing the power of user-defined spectral intervals or (O2) improving the flatness within a user-defined spectral window. Here, $P_i$ denotes the power in spectral interval $i$, where the indices 'RT' and 'opt' distinguish between room temperature and the optimized temperature pattern, respectively. To calculate $P_i$, the power spectral density (PSD) is integrated over the specified wavelength interval that is defined by the target wavelength $λ_{t,i}$ and the width $Δλ$. Note that in simulations this integration is applied to the PSD, while in experiments the integrated power per wavelength $P_i(λ)$ is considered, since this is the quantity measured by the optical spectrum analyzer (OSA). The average power of the interval $i$ is represented by $\overline{P}$.

$$P_i^{sim} = \int_{λ_c - λ/2}^{λ_c + λ/2} PSD(λ)\, dλ; \qquad P_i^{exp} = \sum_{λ_c - λ/2}^{λ_c + λ/2} P_i(λ) \qquad (1)$$

The optimization objectives (O1 and O2) are considered separately in simulations and experiments to fully demonstrate the potential of our concept. Note that the simulations are designed to

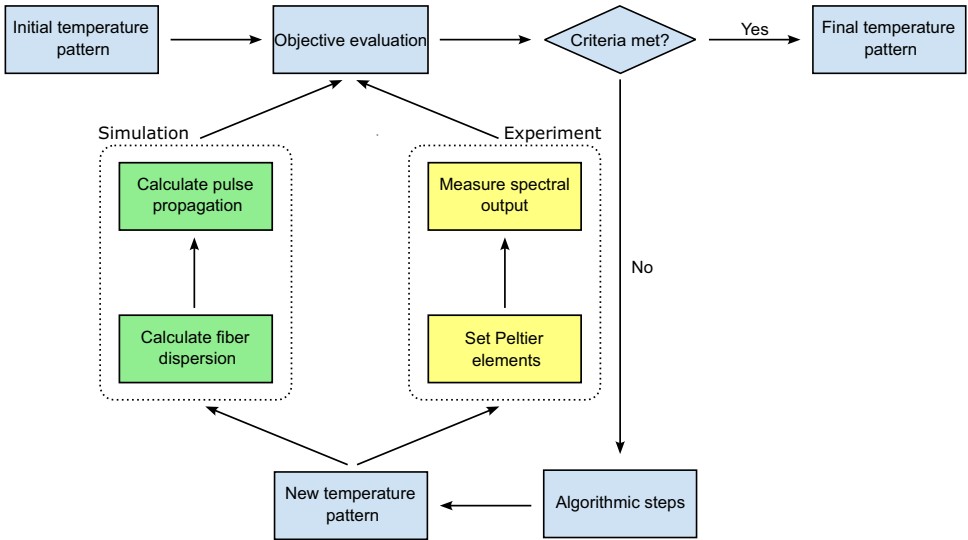

**Fig. 3 | Schematic of the optimization process including objective evaluation, algorithmic steps, and generation of new solutions.** Note that the process differs between simulations and experiments. In simulations, the new temperature pattern is used to calculate the modified fiber dispersion and pulse propagation, while in experiments, the new temperature pattern is implemented via Peltier elements to measure the resulting output spectrum.

highlight the system's capabilities rather than to precisely replicate experimental conditions. The simulations make unrestricted use of the parameter space, assuming precise temperature control, absence of thermal disturbances, and no initial unmodulated fiber section, thereby allowing maximum flexibility in all relevant parameters without introducing constraints from the specific setup used here. This strategy ensures that simulations and experiments complement each other effectively, while experimental optimization is conducted independently of the specific LCF configuration, enabling optimized output spectra even when the fiber parameters are not precisely known.

**O1: Optimization of power in selected spectral intervals.** The first optimization objective is to maximize the spectral power $P_i$ in two independently defined spectral intervals, which leads to the following definition of the objective function O1:

$$f_{O1} = \sum_i \frac{1}{c_i P_i}. \quad (2)$$

The objective function sums the inverse powers $P_i$ of each interval, generally weighted by factors $c_i$, which are set to 1 in this study. Minimization of this sum is required to achieve the optimization goal. Note that by summing the inverses rather than taking the inverse of the sum or negating the sum, the effect of a single dominant interval on the objective function is avoided. Due to its simplicity, this approach is preferable to introducing penalty terms. To quantify the optimization result, the enhancement factor $\eta_i$ of the spectral interval $i$ and the mean enhancement factor $\overline{\eta}$ of the entire optimization are defined with respect to the ratio of the optimized power (indexed 'opt') to the power at room temperature (indexed 'RT'):

$$\eta_i = \frac{P_{i,\text{opt}}}{P_{i,\text{RT}}}; \qquad \overline{\eta} = \frac{\sum_i^N \eta_i}{N}. \quad (3)$$

In addition, the power contrast $C$ between two intervals is defined to highlight the difference in the final power of the intervals:

$$C = \frac{P_1 - P_2}{P_1 + P_2}. \quad (4)$$

**O2: Generation of a flat spectrum in a defined spectral interval.** The second optimization objective (O2) is to achieve a flat spectral power distribution within a specified spectral region, which leads to the definition of the following objective function:

$$f_{O2} = \frac{1}{\overline{P}} \sqrt{\frac{1}{\lambda} \int_{\lambda_c - \lambda/2}^{\lambda_c + \lambda/2} (P - \overline{P})^2 \, d\lambda}. \quad (5)$$

Analogous to the RMS roughness, flatness is defined as the relative standard deviation within the selected interval, with a smaller value (approaching 0) indicating a flatter distribution. This definition also establishes the later used flatness parameter $F$. Note that the relative standard deviation is used to focus only on relative deviations or irregularities. This approach ensures comparability between spectra of different mean power levels and focuses only on relative variations.

### Simulations
In the following, the concept of computationally optimized nonlinear frequency conversion using pLCF is investigated using simulation and optimization procedures described above.

**O1: Optimization of power in two intervals.** To demonstrate the capabilities of the chosen approach for spectral power optimization in selected spectral regions, Fig. 4 shows an example of an optimization result with two intervals in the near-IR region ($\Delta\lambda = 50$ nm, $\lambda_{t,1} = 1300$ nm, $\lambda_{t,2} = 2200$ nm) according to the defined objective function Eq. (2). A significant increase and concentration of power is found in the target spectral intervals (Fig. 4a, intervals highlighted in light red, solid line) compared to the room temperature scenario (dashed line), achieving an average enhancement factor of $\eta = 59$ and a contrast of $C = -0.05$ ($\eta_1 = 22$, $\eta_2 = 96$). Two more examples including a 3 interval optimization can be found in the Supplementary Note 3.

To highlight the importance of algorithmic optimization, Fig. 4b shows the spatio-spectral pulse evolution, revealing complex dynamics driven by the non-intuitive spatial variations of ZDW and GVD. Figure 4c shows the corresponding temperature distribution that induces the obtained complex ZDW and GVD evolution along the fiber needed for the optimization. Such a complex target distribution, resulting in complex non-linear interactions neither allows to isolate individual broadening effects nor their interplay. In general, the

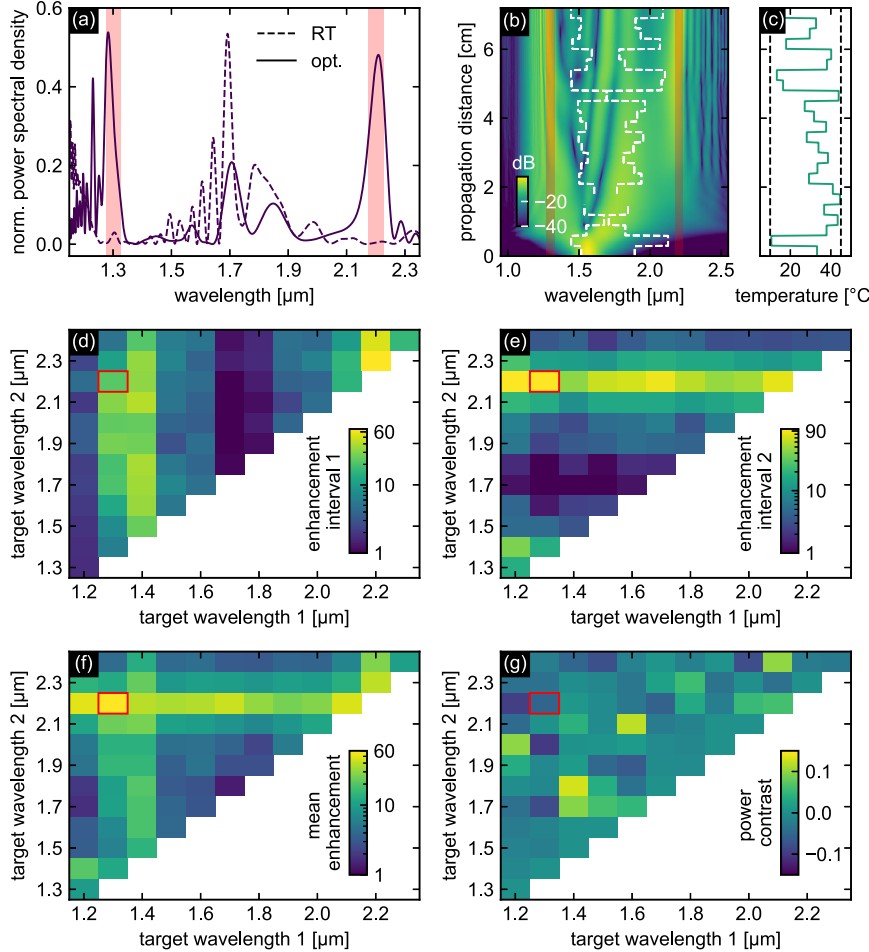

**Fig. 4 | Simulation results of the PSO to maximize the spectral power in two independent spectral intervals according to Eq. (2). a** Example ($\Delta\lambda = 50$ nm, $\lambda_{t,1} = 1300$ nm, $\lambda_{t,2} = 2200$ nm) of an optimized output spectrum (solid line) compared to the configuration at room temperature (dashed line). The light red areas indicate the optimization intervals. **b** Corresponding spatio-spectral pulse evolution using the optimized dispersion distribution. The curves in **a** and **b** are normalized to the maximum power of the input pulse. The white dashed lines indicate the evolution of the ZDWs. **c** Corresponding temperature distribution. The vertical black dashed line indicates the minimum and maximum possible temperatures. **d**, **e** Enhancement factor of interval 1 and 2, respectively, for different combinations of target wavelengths ($\Delta\lambda = 50$ nm). The highlighted tile represents the example shown in **a**. Note that the absence of data in the lower right part of the plot is due to combinatorial reasons. **f** Mean enhancement factor for different combinations of target wavelengths ($\Delta\lambda = 50$ nm). **g** Power contrast showing the difference in the final spectral power of the intervals.

different spectral features occur at different times and positions along the fiber. Abrupt temperature changes induce complex soliton dynamics, including cascaded DW emission or soliton breakdown characterized by sudden DW emission. Furthermore, significant reduction or disappearance of the AD region can trigger soliton reforming after passing through a defined domain of normal dispersion, potentially accompanied by DW emission[20,45].

An overview of the potential of the PSO approach is provided by calculating the enhancement factor for different combinations of target wavelengths with a constant bandwidth $\Delta\lambda = 50$ nm in Fig. 4d. The greatest improvements are observed in regions where the spectral power at room temperature is low, e.g. between 2100 nm and 2300 nm or 1300 nm and 1400 nm. Particularly notable are combinations where one interval lies in each of these spectral ranges (e.g., the example shown in Fig. 4a, $\lambda_{t,1} = 1300$ nm and $\lambda_{t,2} = 2200$ nm). A possible reason for this could be that the fraction of total power that can actually be transferred to other spectral regions is limited on both the short- and long-wavelength sides, with a defined portion always present on either side. Therefore, it is not possible to concentrate the entire spectral power on one side (e.g., complete suppression of DWs on the short-wavelength side in favor of the long-wavelength side). As a result,

optimizing two intervals on the same side of the AD domain is less effective as it requires distributing the available power of this side of the spectrum between the two intervals. Conversely, placing the intervals on different sides allows the power on each side to be concentrated entirely in that interval.

**O2: Optimization of spectral flatness.** The second optimization objective (O2) aims to produce spectrally flat spectra within predefined wavelength ranges, as defined by the corresponding objective function in Eq. (5). Figure 5a shows an example with an exceptionally flat region extending beyond the target range ($F_{opt} = 0.036$, solid line), which is a significant improvement over the spectrum of the unmodulated fiber at room temperature ($F_{RT} = 0.987$, dashed line). Notably, the DW at 2.45 $\mu$m in the room temperature case, which consistently appears as a strong feature in the output spectra, is flattened and no longer visible after optimization. Note that similar to the previous case O1, the non-intuitive temperature distribution (Fig. 5c) and the associated GVD result in highly complex soliton dynamics (Fig. 5b), driven by the nonlinear frequency conversion processes mentioned earlier. One more example can be found in the Supplementary Note 4.

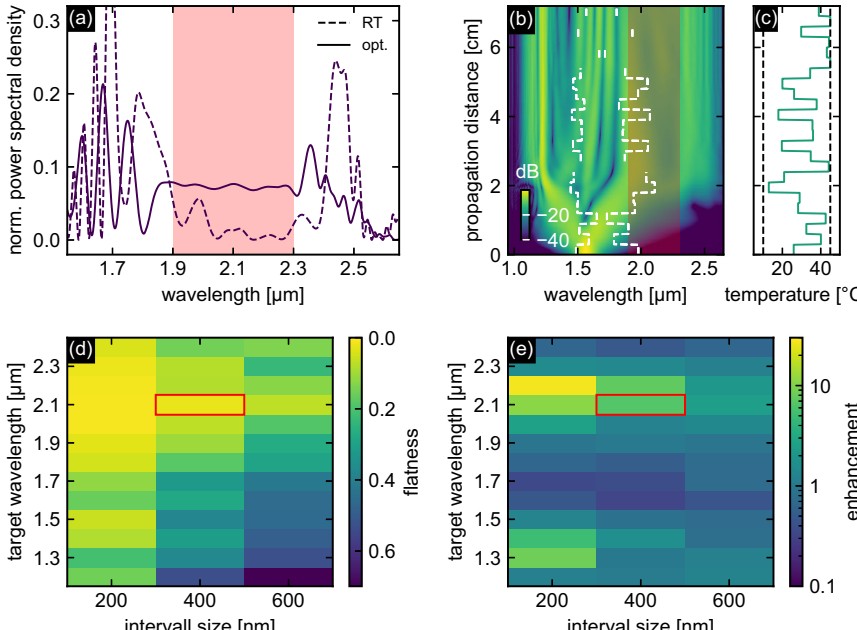

**Fig. 5 | Simulation results of PSO targeting flat spectral output at high power levels according to Eq. (5). a** Example ($\Delta\lambda = 400$ nm, $\lambda_t = 2100$ nm) of an optimized output spectrum (solid line) compared to the unoptimized configuration at room temperature (dashed line). The light red area refers to the predefined optimization interval. **b** Corresponding spatio-spectral pulse evolution. The white dashed lines indicate the evolution of the ZDWs. The curves are normalized to the maximum power of the input pulse. **c** Corresponding temperature distribution. The vertical black dashed line indicates the minimum and maximum possible temperatures. **d**, **e** Flatness factor and enhancement factor for different combinations of target wavelength and bandwidth.

The ability of the PSO approach to produce spectrally flat SCG spectra within predefined spectral domains is demonstrated by the flatness parameter $F$ as in Eq. (5) for various combinations of target wavelength and bandwidth (Fig. 5d). In general, most combinations result in relatively flat spectra, with less flat spectra observed only at very short target wavelengths or for exceptionally large interval widths. In addition, Fig. 5e illustrates the simultaneous power enhancement for the various parameter combinations, which was considered as a secondary optimization objective but not explicitly included in the objective function. Spectral regions of high initial power, such as pump, soliton, and DWs, show spectral flattening accompanied by negligible power enhancement. In contrast, regions of low initial power allow the generation of spectra that are both flattened and enhanced in mean power.

**Experimental validation**

To experimentally demonstrate the capabilities of the method, a temperature control setup was used in combination with a $CS_2$-filled pLCF, targeting multi-wavelength and flatness optimization considering the two defined objective functions $f_{O1}$ and $f_{O2}$. An array of 24 individually addressable Peltier elements of length 2.8 mm is in thermal contact with the pLCF, leading to a total length of the temperature-modulated section of approximately 7.2 cm. The relevant temperature range in experiments and simulations is 10 °C to 45 °C, limited by technical restrictions and the boiling point of $CS_2$. A thermal camera (FLIR A70 29°) is used to visualize the final temperature profile, but is not part of the actual optimization process.

For autonomous local dispersion control, the feedback loop includes a spectrometer, electronics, and Peltier elements. A computer acts as the central unit, processing the measurement data, evaluating the results, and sending instructions to the microcontroller, which adjusts the spectral output based on the temperature changes induced by the Peltier elements. The fiber is pumped by a laser providing 30 fs pulses with central wavelength 1.56 $\mu$m and an average power of 100 mW. For details about the fiber and optical setup, see Methods.

**O1: Optimization of power in two intervals.** The first experimental demonstration focuses on optimization in two predefined spectral intervals defined by the objective function $f_{O1}$. To proof the functionality, Figure 6 shows four examples of spectral power optimization in various combinations of selected intervals, targeting short and long wavelength spectral ranges. In all cases, a significant increase in spectral power is observed in both intervals. The key performance metrics are summarized in Table 1. The corresponding temperature distribution of the Peltier elements (insets in Fig. 6a–d) is of particular interest, which show non-intuitive profiles that would not have been achievable without algorithmic optimization. This result clearly demonstrates the effectiveness of the system in enhancing power in selected spectral intervals and highlights the essential role of computational optimization in directing power into predefined spectral domains. It should be noted that room temperature spectra can vary between different optimization runs, resulting in different optimized spectra despite identical optimization parameters and objectives. This variation is mainly due to different coupling conditions as the measurements were performed on different days. It is important to note that these variations are not due to degradation of the LCF or the core liquid, as recent studies have shown stable supercontinuum generation in a $CS_2$-LCF operating in a higher-order mode over many hours[46], as well as stable operation over several days under exposure to very high average laser powers (> 1 W[47]). The variation of the setup over the course of a measurement is minimal, ensured by the advanced setup design and in particular by the piezo-stabilized coupling. This stability has been confirmed by control spectra recorded at room temperature at the end of each iteration by switching off the Peltier elements, which show sufficient stability over the duration of the experiments (see Supplementary Note 5), ensuring reliable optimization throughout the measurement period.

**O2: Optimization of spectral flatness.** To experimentally demonstrate the ability to produce flat output spectra, i.e. to optimize supercontinua with respect to spectral flatness defined by the

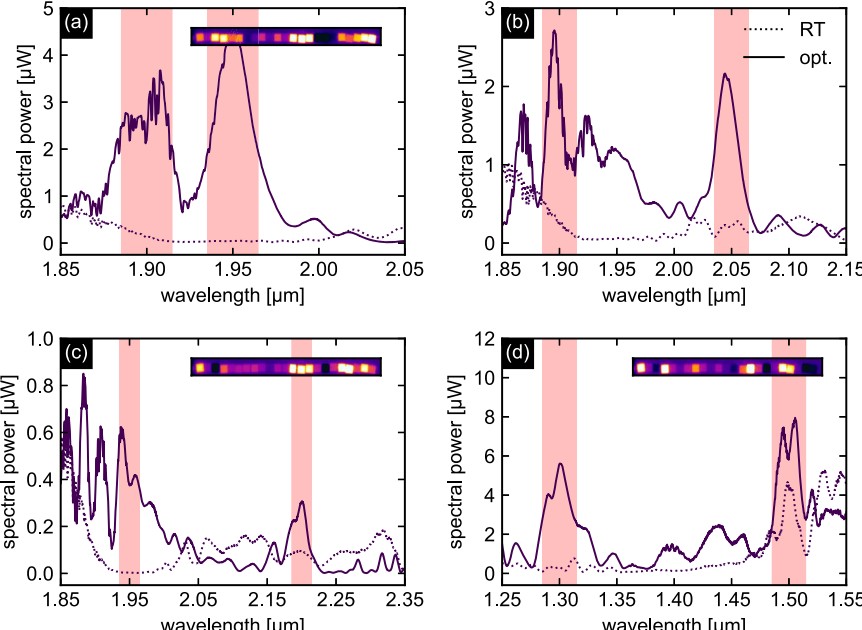

**Fig. 6 | Four examples of experimental optimized output spectra aiming to maximize the spectral power in two selected intervals according to Eq. (2).** **a** $\lambda_{t,1} = 1.90\,\mu m$, $\lambda_{t,2} = 1.95\,\mu m$, **b** $\lambda_{t,1} = 1.90\,\mu m$, $\lambda_{t,2} = 2.05\,\mu m$, **c** $\lambda_{t,1} = 1.95\,\mu m$, $\lambda_{t,2} = 2.20\,\mu m$, **d** $\lambda_{t,1} = 1.30\,\mu m$, $\lambda_{t,2} = 1.50\,\mu m$. All examples consider $\Delta\lambda = 30$ nm. The solid lines refer to the optimized spectra, the dashed lines to the corresponding room temperature spectra. The light red areas indicate the optimization intervals. The thermal images inserted into each plot show the measured temperature distribution of the Peltier elements along the fiber. Note that the image file related to **b** is corrupted and therefore cannot be shown.

**Table 1 | Key performance metrics for the four examples (see Fig. 6) of experimental maximizing of the spectral power in two selected wavelength intervals with a bandwidth $\Delta\lambda = 30$ nm according to Eq. (2)**

|  | $\lambda_{t,i}$ [$\mu m$] | $\eta_1$ | $\eta_2$ | $\bar{\eta}$ | $C$ |
|---|---|---|---|---|---|
| Example (a) | 1.90; 1.95 | 17.8 | 78.5 | 48.1 | −0.11 |
| Example (b) | 1.90; 2.05 | 5.8 | 7.5 | 6.7 | 0.04 |
| Example (c) | 1.95; 2.20 | 153.1 | 2.6 | 77.8 | 0.35 |
| Example (d) | 1.30; 1.50 | 11.8 | 2.0 | 6.9 | −0.16 |

objective function $f_{O2}$, two representative examples of optimized spectra in the short wavelength spectral range are shown in Fig. 7 ((a) $\lambda_t = 1300$ nm, $\Delta\lambda = 200$ nm; (b) $\lambda_t = 1400$ nm, $\Delta\lambda = 200$ nm). After optimization, flatness values of $F_{opt} = 0.57$ and $F_{opt} = 0.61$ were achieved, representing a significant improvement compared to the corresponding room temperature spectra values of $F_{RT} = 2.38$ and $F_{RT} = 1.85$, respectively. The improved flatness was accompanied by an increase in total spectral power within the intervals. All relevant metrics showing the improvement are summarized in Table 2. As seen earlier, the optimized spectra are driven by non-intuitive temperature distributions (insets in Fig. 7), highlighting the essential role and effectiveness of algorithmic optimization in achieving the desired spectral properties.

## Discussion
### Improvement of experimental configuration
A challenge of the present experimental setup is the relatively long measurement time, primarily caused by the slow transition to thermodynamic equilibrium of 7 seconds on average. The slow time response is primarily due to the properties of the Peltier elements used, which have relatively large thermal mass and inertia. An alternative solution is to employ thermal print heads, commonly used in applications such as barcode and receipt printing, which offer response times in the range of 1 to 5 ms[48]. This improvement results from their lower thermal mass and direct resistive heating mechanism, as opposed to the thermoelectric operation of Peltier devices. Metallic wires similarly enable rapid heating because of their low thermal mass. Furthermore, integrating microfabricated resistive microheater arrays - such as metal or ceramic thin films - directly onto the fiber surface could enable fast and highly localized temperature modulation, potentially on millisecond time scales (e.g. a Si-based thin-film microheater with a thermal rise time under 20 ms was demonstrated in Ref. 49). In addition, light-to-heat approaches to generate temperature distributions may also help to shorten relaxation times. Recent studies have shown that coating LCF with carbon nanotubes allows modulation of the supercontinuum radiation by external light irradiation on timescales of a few seconds (e.g., 1 s response time was demonstrated in Ref. 38). In principle, this approach could be further accelerated by introducing light-absorbing materials (e.g. graphene) directly into the liquid core. Note that reducing the optimization time would also allow quantification of the uncertainty of the measured optimization performance (Tab. 1 and 2) through a detailed statistical analysis, which is currently not feasible due to the comparably long duration of the optimization.

In this study, the spatial resolution of the temperature distribution is defined by the width of the Peltier elements (2.8 mm), which is sufficient for the current experimental configuration, i.e., the specific combination of LCF parameters and pulse properties. If required, higher resolution could be achieved with alternative methods, like thermal printer heating elements, potentially extending applicability to effects requiring longer pulses, such as quasi-phase-matched harmonic generation. Several critical experimental parameters cannot be fully incorporated into the simulations, making a direct comparison between experimental and simulated results challenging. These include temperature variations from the ideal distribution due to imperfect thermal coupling between the fiber and Peltier elements (c.f.

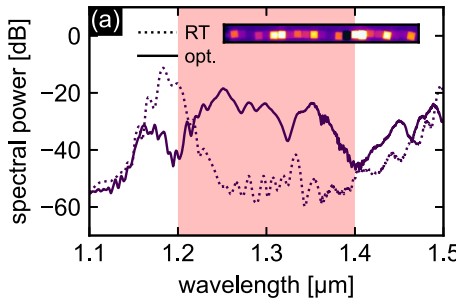 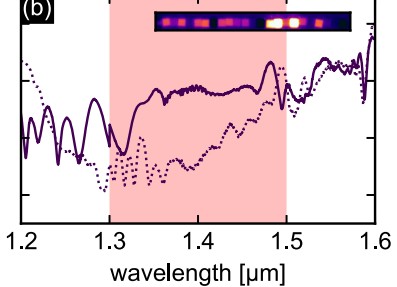

**Fig. 7 | Two examples of experimental optimized output spectra aiming to achieve flat spectral output in a selected interval with bandwidth $\Delta\lambda = 200$ nm. a** $\lambda_t = 1300$ nm, **b** $\lambda_t = 1400$ nm. The solid lines refer to the optimized spectra, and the dashed lines refer to the corresponding room temperature spectra. The light red area indicates the optimization interval. The insets show the measured temperature distributions provided by the sequence of Peltier elements.

**Table 2 | Key performance metrics for the experimental optimization examples (see Fig. 7) for generating flat output spectra combined with increased spectral power within the selected intervals with a bandwidth $\Delta\lambda = 200$ nm according to Eq. (5)**

|  | $\lambda_t$ [$\mu$m] | $\eta$ | $F_{opt}$ | $F_{RT}$ |
|---|---|---|---|---|
| Example (a) | 1.30 | 4.9 | 0.57 | 2.38 |
| Example (b) | 1.40 | 3.5 | 0.61 | 1.85 |

Supplementary Note 6), the presence of unmodulated fiber sections at the input and output in the experiment, and additional uncertainties such as assuming an idealized input pulse, unknown actual power levels in the fiber, and possible cross-talk between Peltier elements.

## Choice of objective functions
A key element of the PSO-approach is the definition of objective functions. Although the suitability of the objective functions used here has been demonstrated experimentally and through simulation, other objective functions could in principle give better results. Possible improvements could include adjusting the definition of the existing objective functions, redefining the flatness metric (e.g., using the ratio of geometric to arithmetic mean), fine-tuning the final spectrum by adding penalty terms, and more. In addition, future optimizations of the PSO hyperparameters may allow for more tailored and efficient optimization strategies.

## Alternative optimization algorithms
The PSO algorithm used in this study provides an efficient solution for the 24 Peltier element configuration. However, future research should investigate whether alternative algorithms could provide better and faster optimization results. In addition to particle swarm optimization, reinforcement learning (RL) offers a promising alternative as it not only finds optimal states but is also more robust to system changes. While PSO only works ideally for fixed environmental conditions, RL can continuously respond to changes in environmental parameters and adjust temperature control to achieve and maintain optimal operating conditions. This is in contrast to PSO, which requires re-optimization when changes occur. Studies show the successful use of RL for self-tuning of optical systems, robust control despite noise and automated system optimization[50–52]. These approaches could open up new perspectives for liquid core fibers, especially for the dynamic adaptation of temperature control under fluctuating environmental conditions. In addition, a combination with numerical methods for the unsupervised identification of dominant physical processes is conceivable. This would enable the automatic detection of locally acting nonlinear processes and facilitate their targeted optimization[53].

## Machine Learning
In addition to optimizing output spectra, the presented pLCFs can can be transferred to other areas, especially towards optical information processing and neuromorphic computing. Such temperature-driven control can enable a proposed training concept for waveguide-based computing approaches[54]. Indeed, the use of nonlinear pulse dynamics in multimode fibers[55] and spectral fission in single-mode fibers[56] have attracted increasing interest in recent years for efficient, scalable optical computing.

## Comparison of reconfigurable supercontinuum approaches
Another approach to reconfigurable ultrafast SCG involves pulse shaping techniques that adjust the input pulse characteristics to influence nonlinear dynamics without modifying the fiber itself[40–42,57]. While this method allows for rapid reconfigurability, it is inherently limited by the static dispersion properties of the fiber, which can limit the extent of spectral modulation. In contrast, the approach presented in this study directly optimizes the fiber properties, providing greater flexibility in shaping dispersion landscapes. In addition, pulse shaping often requires complex and expensive equipment, whereas the proposed method provides a more accessible and straightforward control framework. Note that although local dispersion tuning is a powerful tool, it is fundamentally limited by physical constraints such as phase matching and soliton dynamics, as well as the inability to produce abrupt, strong GVD changes. Thus, combining input pulse shaping with local GVD control may allow access to a broader range of seed events and four-wave mixing processes, potentially enabling on-demand spectral shaping across the full bandwidth of a target supercontinuum. Mechanical deformation of multimode fibers provides an alternative approach that uses controlled bending to adjust dispersion characteristics and redistribute spectral power[4,43]. Although this method allows for dispersion modulation, its flexibility and tuning range are inherently limited, and multimode operation may not be suitable for certain applications. In addition, mechanical adjustments can present challenges such as potential fiber damage and reduced dispersion control accuracy. Note that the use of a higher-order mode in this work does not limit the application range of the LCF platform, as recent advances in 3D nanoprinting enable the fabrication of efficient on-fiber mode converters - such as phase-only holograms[58] or metasurfaces[59] - that can readily convert higher-order modes to fundamental modes.

## Photonic integration
The pLCFs used in this study have significant potential for photonic integration, as their concentric step-index refractive index profile allows them to be spliced and integrated into fiber circuits. Recent

work has successfully demonstrated such integration in the context of stimulated Brillouin-Mandelstam scattering, highlighting their applicability in advanced fiber-based photonic systems[60]. Note that carbon chlorides[61] may represent alternative candidates for implementing programmable fibers, as they exhibit high thermo-optic coefficients and a smaller refractive-index contrast with silica. Another relevant type of waveguide are hybrid fibers containing unconventional materials[62], which can also be used for temperature modulation in case of suitable core materials. Examples of such hybrid waveguides include silicon or chalcogenide core fibers[63–67], which in principle offer very high thermo-optic coefficients along with strong nonlinearities, making them potentially relevant for programmable fibers and temperature-sensitive waveguide applications. A detailed comparison of the properties of materials with high thermo-optic coefficients is provided in Supplementary Note 7. Note that it has recently been shown that LCFs with microstructured claddings can be fabricated to potentially enhance temperature sensitivity[68], though further investigation is needed to determine whether tailored external temperature patterns can be efficiently transferred to the fiber core and if the temperature sensitivity of the guided modes - ideally the fundamental mode - can be further improved by using a microstructured cladding.

### Future experiments

The PSO-based optimization approach is not limited to SCG, but can also be applied to other nonlinear effects and waveguide systems[69], such as higher harmonic generation or multimode fibers. Since quasi-phase-matched higher-order DW generation has recently been demonstrated in LCFs with periodically modulated core diameters[22], the programmable fiber platform presented here could provide a valuable means to study this highly dispersion-sensitive phenomenon in a reconfigurable manner. Moreover, preliminary simulations of LCF configurations operating entirely in the normal dispersion regime show that the output spectra can be optimized for flatness in these cases, though further studies are needed to fully explore the capabilities and limitations. Since the modulation mechanism is based on local changes in the waveguide dispersion, the method is also applicable in linear photonics, e.g. for the optimization of fiber-integrated spectral filters or tunable dispersion compensation elements.

In this work, we have introduced the concept of computationally-optimized nonlinear frequency conversion in programmable liquid-core fibers that enables real-time tunable and reconfigurable nonlinear power distribution through computationally-optimized dispersion landscapes. Central to this concept is the integration of a temperature-sensitive mode in a liquid-core fiber, soliton fission-based super-continuum generation, particle swarm optimization, and a computer-controlled heating array, forming a sophisticated feedback loop for precise spectral control of ultra-fast supercontinuum generation. This combination allows algorithmic optimization of the supercontinuum output through local temperature-induced dispersion modulation along the pLCF. Significant improvements in spectral power density in multiple spectral intervals simultaneously and enhanced spectral flatness have been demonstrated both experimentally and in simulations, underscoring the robustness and adaptability of the platform. In particular, the experimental validation confirms the feasibility of implementing computationally optimized dispersion profiles in real fiber systems using a framework that relies exclusively on experimentally accessible parameters, paving the way for new advances in programmable light control in nonlinear photonic systems. From an experimental perspective, the optimization is performed independently of the specific LCF configuration, allowing optimized output spectra to be achieved even when the fiber parameters are not precisely known.

The presented approach introduces a new paradigm for the design of reconfigurable nonlinear frequency conversion and shows a remarkable versatility that extends far beyond supercontinuum generation. It can be effectively applied to a wide range of linear and nonlinear phenomena, including adaptive harmonic generation, tunable soliton dynamics, spectral filtering, and can be used in multimode and hybrid fiber and waveguide systems. This adaptability and wide range of potential applications create exciting opportunities for both fundamental research and practical implementations demanding advanced photonic control.

## Methods

### Fiber design and preparation

The pLCFs used in this study consist of a 13 cm long, in-house fabricated, fiber-like silica capillary with an inner diameter of 3.92 $\mu m$, which is filled with $CS_2$ by capillary action within minutes. Note that although $CS_2$ is toxic in large quantities, the volume used in the LCF is extremely small (e.g., only 1.2 nL for the 10 cm LCF discussed here), rendering toxicity concerns negligible when the fiber is integrated into a spliced environment[60]. The core diameter was chosen to ensure that the radially polarized $TM_{01}$-mode exhibits two ZDWs at room temperature to position the central pulse wavelength $\lambda_p = 1560$ nm in the AD region, thus allowing soliton-based effects (c.f. [44] for details). Note that when considering short fibers (< 20 cm), the high transparency of $CS_2$ enables spectral operation from 400 nm to approximately 4 $\mu m$[70,71]. Optical and fluid access was provided by using optofluidic mounts at the input and output of the fiber. These mounts have refillable liquid reservoirs accessible via ports and provide optical access through transparent windows (see ref. [72]).

### Optical setup

The optical setup for the computational optimization includes an ultra-fast laser, excitation optics, pLCF, array of heating elements, electronic control, diagnostics, and a computer-controlled optimization feedback loop (see Supplementary Note 8). Pulses from the ultrashort laser with the central wavelength 1560 nm, the pulse duration 30 fs and the repetition rate 80 MHz (Toptica FemtoFiber Pro II) are coupled into the pLCF via an aspherical lens (L1, Thorlabs C230-C). Laser power is adjusted using a combination of a half-wave plate (HWP) and linear polarizer (LP). The linearly polarized input light is converted into the $TM_{01}$ mode using an s-wave plate (details in ref. [44]). The output light is collected either with an aspherical lens (C036TME-D) or a reflective objective (LMM40XF-P1) without chromatic aberration (L2) and coupled into a spectrometer (Yokogawa AQ6375) for spectral analysis. Note that when using the aspherical lens, the impact of chromatic aberrations was minimized by optimizing the coupling into the OSA to the relevant wavelength domain. As higher-order mode SCG is highly sensitive to experimental conditions, input coupling was stabilized against mechanical drift using a closed-loop piezo-controlled stage (Thorlabs MAX311D/M). With this setup, a stable coupling efficiency of 20% into the $TM_{01}$ mode (ratio of power before and after the lens, uncorrected) was achieved over several hours at an average input power of $P_{avg} = 100$ mW (for details see ref. [46]).

### Thermal control unit

The used Peltier elements for the local dispersion control were MTEC-0.4-0.3-0.07-71-3/4 by 'Arctic TEC Technologies' with a length of 2.8 mm. An array of 24 individually leading to a total length of approximately 7.2 cm was attached to a metal block that acts as heat sink and thermally contacted to the pLCF. The temperature is electronically controlled by a pulse-width modulated (PWM) signal. The pulse width modulated signal is provided by an Arduino Mega, driving an additional H-bridge (L9110S) allowing polarity reversal for both heating and cooling. Subsequently the voltage signal is smoothed and limited by resistors and capacitors leading to the required temperature range of 10 °C to 45 °C. Note that the densely mounted elements cause unavoidable thermal interference that limits the creation of arbitrary temperature profiles with large steps.

To reduce the time required to reach thermal equilibrium after the temperature distribution is set, a dynamic control method is used: Initially, the PWM signal exceeds the target duty cycle to accelerate heating or cooling of the Peltier elements, followed by gradual adjustment. This dynamically adjusts the delay time based on the difference between successive patterns. The resulting temperature control times ranged from a theoretical minimum of 0 s to a maximum of 25 s in exceptional cases, with an average of 7 s observed in real experiments. With the presented temperature-control unit and the assumed need of 7 s per temperature pattern a single iteration with a swarm size of 50 particles takes about 6 min. Hence, dominating the needed time of the optimization process compared to spectral acquisition and evaluation.

### Fiber dispersion calculations

The calculation of the fiber dispersion and the effective mode area were performed using the Python package fibermodes[73], enabling precise modeling of temperature-dependent optical properties. The temperature dependent behavior of the $TM_{01}$ mode in the $CS_2$-filled LCF with a core diameter of 3.92 $\mu$m is included using the temperature dependent Sellmeier equation for $CS_2$[35].

### Nonlinear pulse propagation

Simulations of nonlinear pulse propagation and calculation of output spectra were performed by solving the Generalized Nonlinear Schrödinger Equation (GNLSE) using a modified version of the Python package gnlse-python[74] using the previously received fiber dispersion. The input pulse parameters (sech-square shape, pulse duration 30 fs, center wavelength 1560 nm, peak power 10 kW) remain constant throughout the simulations. Unlike the experiment, the simulations consider a 7.2 cm long fiber (equivalent to 24 TECs) that is fully temperature modulated.

### Particle Swarm Optimization

A PSO according to[75] was implemented using the PySwarms python library[76] without the use of further modifications such as adaptive hyperparameters. No exhaustive search for optimal algorithm parameters was performed. Instead, parameters were chosen based on guidelines and previous experience. The used parameters are listed in Supplementary Note 9.

## Data availability

The presented simulated and measured spectral data generated in this study have been deposited in the Zenodo database under accession code 10.5281/zenodo.16811141[77].

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

## Acknowledgements

We acknowledge support from the German Research Foundation (Deutsche Forschungsgemeinschaft, DFG) via the grants QI140/2-1, SCHM2655/21-1, SCHM2655/23-1, SCHM2655/3-2, JU3230/1-1. B.F. acknowledges funding by the Klaus Tschira Boost Fund (KT51). M.C. acknowledges funding by Carl-Zeiss-Stiftung via the Nexus program (project P2021-05-025). We acknowledge support by the German Research Foundation Projekt-Nr. 512648189 and the Open Access Publication Fund of the Thueringer Universitaets- und Landesbibliothek Jena.

## Author contributions

J.H. conceived and conducted the experiments and simulations. R.S. supported experiments and contributed to the scientific discussion. B.F, M.C. gave support and advice to the algorithmic approach. M.C. contributed to the scientific discussion. M.A.S supervised the work. J.H. and M.A.S. prepared the manuscript with contributions from all authors. All authors revised and commented on the manuscript.

## Funding

## Competing interests

All authors are listed as inventors (Johannes Hofmann, Ramona Scheibinger, Bennet Fischer, Mario Chemnitz, Markus A. Schmidt) on a pending patent application (Application No. DE 10 2025 103 337.4) filed by Leibniz-IPHT (non-profit). The patent application is currently pending at the German Patent and Trademark Office (Deutsches Patent- und Markenamt, Germany). It covers a system and method for thermal mode control and autonomous generation of user-defined optical fields in fibers.
