## [Transparent Peer Review file · Nature Communications]

Programmable Liquid-Core Fibers: Reconfigurable Local Dispersion Control for Computationally Optimized Ultrafast Supercontinuum Generation

Corresponding Author: Professor Markus Schmidt

Version 0:

Reviewer comments:

Reviewer #1

(Remarks to the Author)

The authors demonstrate how the supercontinuum spectrum obtainable from a capillary filled with carbon disulfide can be optimized by controlling the temperature distribution (and therefore the dispersion) along the capillary. The optimization is implemented through the use of a PSO (Particle Swarm Optimization) algorithm: the theoretical analysis is based on spectrum calculations using the GNLSE (Generalized Nonlinear Schrödinger Equation), and the experimental realization involves the local temperature control of a 7.2 cm long capillary using a sequence of 2.8 mm long Peltier modules. Numerical analysis shows that by optimizing the temperature distribution, it is possible to either maximize the power of the supercontinuum in selected spectral sub-bands or achieve a flat spectral power density over a bandwidth of several hundred nanometers. Experimental results confirm that by optimizing the temperatures of the Peltier elements, it is possible to enhance the power in two 30 nm intervals or to obtain a flat spectrum over a 200 nm range. It is noted that the capillary mode considered is TM₀₁, and that the analysis of supercontinuum generation in liquid-filled capillaries (also known as liquid core fibers, LCFs) is a topic that has been extensively studied in recent years and to which the authors have contributed, as evidenced by the reference list.

The explanations provided in the manuscript are very clear and rich in detail, especially in the Methods section, and are undoubtedly of great interest to all researchers working on nonlinear effects in liquid core fibers and optical fibers in general. In my opinion, the work presents only one flaw that requires further explanation and additional analysis from the authors: all theoretical profiles of the ZDWs (Zero Dispersion Wavelengths) along the capillaries show step-like behavior corresponding to the different temperatures chosen for the 2.8 mm Peltier elements, and these ZDW steps should correspond to abrupt temperature jumps. While such jumps are achievable on the external surface of the capillary in contact with the Peltier elements, they likely cannot exist in the internal region of the capillary filled with carbon disulfide. It seems implausible that the temperature distribution inside the capillary could feature such sharp variations, and the infrared camera images do not provide significant information since they refer only to the external surface temperature of the capillary. I believe the authors should attempt to estimate the capillary internal temperature distribution and verify how it differs from the temperature on the external surface. Of course, the capillary wall thickness and the extent of the contact area between the capillary and the Peltier modules could play an important role.

Since the experiments show that the optimization works, I believe the simulations should also confirm that a step-like temperature distribution is not strictly necessary to maximize the power in two sub-bands or to achieve a flat spectrum over a 200 nm range.

Moreover, to make the manuscript more convincing, the authors should add at least one example of optimization (e.g., to achieve a flat spectrum over a specific interval) obtained both numerically and experimentally, and compare the spectra predicted by the numerical simulator with the measured ones.

In my opinion, a revised version of the manuscript incorporating the above suggestions would fully meet the requirements for acceptance by Nature Communications.

Reviewer #2

(Remarks to the Author)

Review report for "Programmable Liquid-Core Fibers: Reconfigurable Local Dispersion Control for Computationally Optimized Ultrafast Supercontinuum Generation"

In the manuscript, authors presented a temperature sensitive liquid core fiber that combines with a heating array to control the mode dispersion segment by segment along the fiber, therefore termed as a programmable fiber with local dispersion control. Such a fiber opens the opportunity of computational optimization for ultrafast nonlinear interactions along the fiber, particularly the supercontinuum process that highly relies on the dispersion environment to tailor the overall spectral envelope. Indeed, authors have demonstrated by means of the particle swarm optimization the enhancement of power spectrum and the improvement of the spectral flatness. More detailed, they have also performed simulations that could confirm the experimental results, and have tested the spectrum stability over a long time period of few hours.

Such results are solid and could firmly support the novelty of the proposed scheme. I can't agree more that such a programmable fiber optic platform can be of broad interest, opening the way to the study of more complex nonlinear and soliton dynamics.

Yet, it would be appreciated if authors could perform a more basic characterization on the specific nonlinear interaction on their programmable liquid core fiber. For example, they argued that the algorithm-led abrupt change of the zero dispersion points along the fiber is essential, which could excite more complex soliton dynamics such as the tunneling effects. But there lacks of a basic characterization or a showing example purely on the tunneling effects. From the simulation, it was still hard to identify the effect since the spectral evolution was very smoothing showing NO clear wavelength conversion via the tunnelling. So I would wonder what if taking a periodical changed ZDW as a standard characteristic to study the nonlinear interaction, which may provide more insights to the complex soliton dynamics.

Another minor comment is on the spectral flattening example. In the presented result, it seems the dispersive waves are still dominating having the strongest spectral intensity. Is it possible to program the fiber in the normal dispersion regime where spectral broadening usually shows decent flatness compared with that in the anomalous dispersion in the soliton regime? The current solution seems only working for sideband spectrum around the pumping wave (and should not be overlapped with the dispersive wave)?

To conclude, I would be supportive to accept this work that are certainly of high novelty and of broad interest, if provided with in-depth and demonstrated understandings of the "non-intuitive" designs.

Reviewer #3

(Remarks to the Author)

The manuscript by Hofmann et al. presents a novel approach for tailoring the shape of a nonlinearly generated fiber supercontinuum spectrum through longitudinal shaping of the core refractive index of the (liquid-filled) fiber by temperature control. A complex longitudinal dispersion map can in this way be realized, and numerically optimized according to specified criteria. The authors demonstrate optimization of either power or spectral flatness in specific wavelength bands, both numerically and experimentally. Some directions for further research are briefly discussed.

The application of a complex spatial temperature profile, and its optimization using contemporary algorithms (swarm optimization) is to my knowledge novel, and could hold significant potential beyond the particular application demonstrated in the paper. The numerical and experimental procedures as described appear sound to me, and the authors make reasonable claims based on the data obtained. I believe that appropriate detail about the simulations and experiments is given for other researchers to attempt a reproduction of the results, but some of my subsequent comments should be taken into account here.

My main concerns about the work and results as presented are the following:

-Comparison between theory and experiment: The authors state that the 'simulations are designed to highlight the system's capabilities rather than to precisely replicate experimental conditions.' This is all very well, and I know from direct personal experience how difficult it can be to reconcile simulations and measurements. However, given that the simulations appear to resemble the experimental layout quite accurately, I am a bit concerned about the total lack of correspondence between simulations and experiments. As a particular example, the authors optimize a flatness factor of 0.036 (lower is better) in the spectral range 1900-2300 nm based on simulations. However, their best experimental result is a flatness factor of 0.26, and in a completely different spectral window, namely 1300-1500 nm. Surely, if the authors were anywhere close to reproducing the simulated result, they would have shown it. This raises the question if the simulations can be trusted at all, and at the very least, a much more extended discussion of their accuracy, and the possible reasons for discrepancies is needed. In particular, I think there should be some direct comparisons between particular experimental spectra, and corresponding best efforts to reproduce them in simulations.

-Reproducibility: The authors state that results for their key metrics show some day-to-day variation, due to drifts and inaccuracies in the optical setup. This should be better quantified. For instance, if one of their eta-parameters is optimized from scratch on different days, what is the magnitude of deviations? How many of the significant figures given in, e.g. Table 1 can we actually trust?

-Flexibility: The authors discuss that the simulations assume step-like temperature changes, whereas in reality this is not possible. Can one derive fundamental limits to the steepness of temperature gradients based on the physical properties of the medium, and reasonable assumptions about the feature sizes of heaters, fiber cores, etc.?

-Response time: The authors claim reconfigurability as a major achievement of the work, but it seems to me that the reconfiguration time is quite slow, ranging from at least 7 seconds if the desired temperature profile is known, to several hours if a reoptimization is needed. This will be a significant restriction of potential use cases. The manuscript contains some discussion of potential improvements, but is it possible to put some numbers on that, e.g. a lower bound on the reconfiguration times obtainable?

-Generality: The proposed approach could in principle find many applications, and open a new sub-field of nonlinear fiber optics. In this sense, publication in a journal like Nature Comm. would seem appropriate. However, in order to optimize the effect of temperature variations the authors have used a particular (albeit common) nonlinear medium, CS₂, known for its high temperature coefficient. They have further optimized the temperature sensitivity through the use of a particular fiber mode, namely the TM₀₁. Now, the use of CS₂ (or any other particular material) invariably implies restrictions in terms of transmission windows, dispersion profiles etc. It is also noteworthy that CS₂ is rather poisonous. The use of the TM₀₁ mode implies issues with in/outcoupling, in particular if a broadband continuum needs to be brought back to a gaussian-like profile. So I think that at least a semi-quantitative discussion of other media is needed to judge the actual impact and generality of the idea. What level of tunability can be expected, e.g. with commonly used infiltration liquids, what can be achieved in the fundamental mode, or other mode types, etc.

Overall, my impression of the paper is fairly positive, but I think that a proper discussion of the above points is needed to allow both readers and editors to judge the reliability and potential impact of the work.

Version 1:

Reviewer comments:

Reviewer #1

(Remarks to the Author)

In their response, the authors show through numerical simulations how a periodic temperature variation can be achieved under boundary conditions compatible with their experiment, in which the temperature inside a liquid core fiber is controlled by Peltier cells arranged periodically. Moreover, the simulations included in the "supplementary information" demonstrate that periodic step-like temperature profiles are also feasible. In their heat diffusion simulations, the authors considered a 2D silica block rather than a silica capillary filled with carbon disulfide, but in my opinion, the results presented are still convincing, as they confirm the periodic distribution of temperature and how it is influenced by the capillary diameter (Fig. S2(a)) and the spacing between the Peltier cells (Fig. S2(b)).

I believe the revised version of the manuscript can be accepted by Nature Communications.

Reviewer #2

(Remarks to the Author)

I appreciate author's effort to make such a comprehensive and detailed reply to all the comments including mines, and I am satisfied with their revisions on the manuscript.

A remaining question is that how to further extend the spectral range of optimization particularly when with a purpose of spectrum flatening? With presented results, the optimization range seems still quite narrowed if compared with the full range of the supercontinuum. It would be better if authors could give a brief discussion on this question in the manuscript (e.g. in the conclusion part).

With that, I would suggest to accept this novel and inspiring work.

Reviewer #3

(Remarks to the Author)

Comment 1: The authors do not attempt to match simulation and experiments for a particular case as I suggested, but I concede that if they really don't know their incoupled power and their pulse shape, it may be a futile effort. I think it is important that these uncertainties are now clearly stated in the manuscript. As a minor point, I do not see why the presence of unheated sections could not have been included in simulations.

Comment 2: I asked about the uncertainty of measured quantities, e.g. data in table 1, in light of the stated day-to-day variations. The authors do not answer this at all. They mostly discuss that the variations are not due to liquid degradation, which is an interesting point in its own right, but not really what I asked about. The authors argue that the uncertainties are not important because ultimately the setup could be fiber-integrated in a more robust way. I will leave it to the editor to consider whether this is an appropriate attitude to experimental data and measurements.

Comment 3: I think the authors have done a good job of answering this concern, which was also raised by other reviewers.

Comment 4: I think the concern is adequately adressed, and that the information added to the manuscript is interesting and helpful.

Comment 5: I think this point is adequately addressed by the additions to both the main text and the supplementary document. In fact, at closer inspection it appears to me that the carbon chlorides may be even better candidates for tunability,

having similar thermo-optic coefficients, but a smaller refractive-index difference to silica.

Response to comments of Reviewer #1

The authors demonstrate how the supercontinuum spectrum obtainable from a capillary filled with carbon disulfide can be optimized by controlling the temperature distribution (and therefore the dispersion) along the capillary. The optimization is implemented through the use of a PSO (Particle Swarm Optimization) algorithm: the theoretical analysis is based on spectrum calculations using the GNLSE (Generalized Nonlinear Schrödinger Equation), and the experimental realization involves the local temperature control of a 7.2 cm long capillary using a sequence of 2.8 mm long Peltier modules. Numerical analysis shows that by optimizing the temperature distribution, it is possible to either maximize the power of the supercontinuum in selected spectral sub-bands or achieve a flat spectral power density over a bandwidth of several hundred nanometers. Experimental results confirm that by optimizing the temperatures of the Peltier elements, it is possible to enhance the power in two 30 nm intervals or to obtain a flat spectrum over a 200 nm range. It is noted that the capillary mode considered is TM₀₁, and that the analysis of supercontinuum generation in liquid-filled capillaries (also known as liquid core fibers, LCFs) is a topic that has been extensively studied in recent years and to which the authors have contributed, as evidenced by the reference list.

The explanations provided in the manuscript are very clear and rich in detail, especially in the Methods section, and are undoubtedly of great interest to all researchers working on nonlinear effects in liquid core fibers and optical fibers in general.

We thank the Reviewer for the thorough review and positive assessment of our manuscript. We appreciate the insightful comments and are committed to addressing each point raised. Below, we provide a detailed, point-by-point response to each comment and hope our answers address the Reviewer's concerns and further support our findings.

In my opinion, the work presents only one flaw that requires further explanation and additional analysis from the authors:

R1.1: all theoretical profiles of the ZDWs (Zero Dispersion Wavelengths) along the capillaries show step-like behavior corresponding to the different temperatures chosen for the 2.8 mm Peltier elements, and these ZDW steps should correspond to abrupt temperature jumps. While such jumps are achievable on the external surface of the capillary in contact with the Peltier elements, they likely cannot exist in the internal region of the capillary filled with carbon disulfide. It seems implausible that the temperature distribution inside the capillary could feature such sharp variations, and the infrared camera images do not provide significant information since they refer only to the external surface temperature of the capillary. I believe the authors should attempt to estimate the capillary internal temperature distribution and verify how it differs from the temperature on the external surface. Of course, the capillary wall thickness and the extent of the contact area between the capillary and the Peltier modules could play an important role.

We thank the Reviewer for raising the issue of the relevance of the step-like temperature changes assumed in our simulations, which was also mentioned by Reviewer #3 in her/his third comment (R3.3). To address this, we first outline the simulation methodology (Sec. A), then analyze the steepness of the temperature gradient in the context of the LCF used in this

study (Sec. B), and finally discuss the influence of geometric and material parameters on the resulting temperature distribution (Sec. C).

A. Description of simulations: To reveal the internal temperature distribution within the LCF sample, we conducted numerical simulations of the steady-state temperature profile in a 2D silica glass block using a Python-based finite-difference scheme. The model discretizes the Laplace equation on a rectangular grid, with periodic boundary conditions applied along the fiber axis (z -direction). The bottom boundary ($x = 0$) is maintained at a spatially periodic temperature profile ($T_0 = 20^\circ\text{C}$, $T_1 = 80^\circ\text{C}$) to simulate external heating, implementing Dirichlet boundary conditions. The top boundary ($x = 125\ \mu\text{m}$, matching the experimental LCF diameter) employs a Robin (mixed) boundary condition to account for convective heat transfer to the surrounding air ($T_{\text{air}} = 20^\circ\text{C}$). The numerical solution is obtained using a custom Python code by assembling and solving the resulting sparse linear system for the interior points, with all boundary conditions directly incorporated. This approach provides a physically realistic model of the thermal behavior of an infinitely extended silica glass block in contact with air on one side, closely mirroring experimental conditions and incorporating the thermal conductivity of silica ($k_{\text{silica}} = 1.31\ \text{W}/(\text{m}\cdot\text{K})$ [link]) as well as the heat transfer coefficient at the silica–air interface ($h_{\text{heat}} = 10\ \text{W}/(\text{m}^2\cdot\text{K})$ [link]).

B. Temperature distribution in the LCF used in this study

Fig. RL1: Simulated temperature distribution within a silica glass block, with periodic Dirichlet boundary conditions applied at the bottom ($x = 0$) and Robin boundary conditions at the top surface ($x = 125\ \mu\text{m}$). To reflect the experimental configuration, the periodicity Λ was set to 5.6 mm, matching twice the length of one Peltier element (2.8 mm). Further simulation details are described in the text. (a) Temperature map in the xz -plane at $y = 0$. (b) Longitudinal temperature profile at half the size of the glass block ($x = 62.5\ \mu\text{m}$, indicated by horizontal dashed green lines in (a)). The horizontal dashed gray lines indicate the maximum and minimum temperatures considered (20°C and 80°C).

The resulting 2D temperature distribution in Fig. RL1(a) demonstrates that the transverse temperature variation along the x -axis is relatively minor, primarily due to the small outer diameter of the fiber ($125\ \mu\text{m}$). This effect becomes even more apparent when examining the longitudinal temperature profile at the midpoint of the block ($x = 62.5\ \mu\text{m}$, Fig. RL1(b)), which shows only slight deviations from the ideal step-like profile at the curve edges.

C. Temperature distribution in different LCF configurations

Impact of materials: To assess the impact of material parameters on the temperature distribution, it is useful to refer to the simulations described in Sec. B, which are fundamentally based on solving the Laplace equation to determine the steady-state temperature profile. These simulations use periodic boundary conditions along the z-axis to represent periodic heating, with Dirichlet and Robin boundary conditions applied at the bottom and top interfaces to simulate heating from below and exposure to an open environment above. The only direct influence of material properties in these simulations arises in the Robin boundary condition at the top, which models convective heat dissipation into air using the thermal conductivity of silica and the heat transfer coefficient at the silica–air interface. As shown in Fig. RL1 (a), the temperature distribution along the x-direction remains largely uniform, with only minimal deviations near the uppermost region of the glass ($x \leq 125\mu\text{m}$), indicating that the impact of material properties is generally minor.

Impact of geometric features: Figure RL2 presents simulated longitudinal temperature distributions at the center of the glass block ($x = 62.5 \mu\text{m}$), corresponding to the location of the fiber core, for different geometric configurations. As shown in Fig. RL2 (a), small block extensions (e.g., $80 \mu\text{m}$, $125 \mu\text{m}$) result in temperature profiles that closely approximate a step-like function, whereas larger extensions lead to smoother profiles due to increased thermal diffusion.

Fig. RL2: Simulated longitudinal temperature distributions at half the extension (along the x-axis, $x = 62.5 \mu\text{m}$) of the glass block, corresponding to the fiber core position (see Sec. A for simulation details). (a) Temperature profiles for different glass block extensions (indicated in the legend in micrometers, $\Lambda = 5.6 \mu\text{m}$). (b) Temperature profiles for different periodicities (pitch values shown in the legend in micrometers). The x-axis is normalized to the respective pitch to allow direct comparison of the profiles.

Figure RL2(b) shows temperature distributions for different periodicities (pitch values), indicating that for comparably large pitches - such as the experimental value of $\Lambda = 5.6 \mu\text{m}$ - the step-like character of the temperature distribution is preserved, and cross-talk between regions of different temperature remains minimal. Noticeable smoothing only appears for much smaller pitch values.

These findings confirm that, under the experimental parameters used ($\Lambda = 5.6 \mu\text{m}$, extension $125 \mu\text{m}$), the temperature distribution remains sharply step-like and cross-talk between heating elements is small, thereby validating the experimental design.

In response to the Reviewer's comment, we have added these sections including the figures to the Supplementary Information (new Sec. 1) and revised the main text accordingly:

... Due to the arrangement of the heating elements used in the experiments (24 Peltier elements, approx. length 2.8 mm each), the GVD changes discretely, resulting in a stepwise variation of the ZDWs, indicated by dashed white lines in Fig. 2. **It should be noted that additional 2D finite-element simulations of the temperature distribution in a comparable silica glass block for various LCF geometries (see Sec. 1 of the Supplementary Information) reveal only minor deviations from a step-like longitudinal temperature profile when using the experimental fiber parameters, supporting the validity of the step-like temperature distribution assumption. ...**

R1.2: Since the experiments show that the optimization works, I believe the simulations should also confirm that a step-like temperature distribution is not strictly necessary to maximize the power in two sub-bands or to achieve a flat spectrum over a 200 nm range.

We thank the Reviewer for the comment regarding the impact of the step-like temperature transitions on the optimization of the supercontinuum spectra. Based on our previous experience with supercontinuum generation in LCFs, we agree with the Reviewer that the detailed shape of the dispersion curve on such small spatial scales is practically irrelevant for the ultrafast supercontinuum generation configuration used here. Moreover, it is important to emphasize that, from an experimental standpoint, the optimization is carried out independently of the specific LCF configuration, enabling optimized output spectra even when the fiber parameters are not precisely known. In the following, we first present a simulation example of output spectra including deviations from the step-like profiles (Sec. A) and then discuss the relevance of the temperature transition in the context of PSO-based optimization (Sec. B).

A. Nonlinear pulse propagation simulations

To evaluate the impact of temperature transition profiles, we performed exemplary nonlinear pulse propagation simulations that incorporate deviations from an ideal step-like transition in temperature, i.e., dispersion. These simulations follow the same approach used in the manuscript (sech² input pulse, 30 fs pulse duration, center wavelength 1560 nm, peak power 10 kW; 7.2 cm fiber length, 3.92 μ m core diameter, TM₀₁ mode). Specifically, we consider several cases with modified temperature distributions compared to the ideal step-like case (Figs. RL3(a–c), top row; purple: step-like profile, cyan: non-step-like profile), resulting in different distributions of the zero-dispersion wavelength (ZDW) along the fiber axis (Figs. RL3(d–f), middle row).

Fig. RL3: Example of nonlinear pulse propagation simulations for different temperature distribution configurations, including deviations from the ideal step-like temperature profile using the same simulation methodology as described in the main text. Top row (a–c): assumed temperature distributions (purple: ideal step-like, cyan: modified profiles). Middle row (d–f): corresponding distributions of the zero-dispersion wavelength (ZDW). Bottom row (g–i): resulting output spectra, with color coding matching the top and middle rows.

It is evident that, regardless of the assumed profile, the output spectra for both step-like and non-step-like configurations show a high degree of overlap (Figs. RL3 (g–i), bottom row), demonstrating that the step-like temperature distribution is well-suited for investigating programmable LCFs in the context of ultrafast nonlinear light generation, at least for the specific combination of LCF and input pulse parameters studied here.

The variations considered in Fig. RL3 are more pronounced than those observed in the thermal distribution simulations (Sec. A), further supporting the validity of the step-like assumption in the nonlinear pulse propagation modeling.

It should be noted, however, that the results in Fig. RL3 represent a single illustrative example, and part of the observed insensitivity may be specific to the chosen temperature profile. To comprehensively assess the impact of shape of the temperature transition region on the output spectra, a full statistical analysis involving a broader set of transition profiles

would be necessary, which is beyond the scope of the present work and may be addressed in a future study.

Therefore, we decided not to include this specific simulation example in the manuscript or Supplementary Information, but extended the main text to address this point accordingly: ... Due to the arrangement of the heating elements used in the experiments (24 Peltier elements, approx. length 2.8 mm each), the GVD changes discretely, resulting in a stepwise variation of the ZDWs, indicated by dashed white lines in Fig. 2. **It should be noted that additional 2D finite-element simulations of the temperature distribution in a comparable silica glass block for various LCF geometries (see Sec. 1 of the Supplementary Information) reveal only minor deviations from a step-like longitudinal temperature profile when using the experimental fiber parameters, supporting the validity of the step-like temperature distribution assumption. Example nonlinear pulse propagation simulations revealed that the generated output spectra are largely insensitive to the specific shape of the temperature transition at the heating element–air interface, although future studies with full statistical analysis of different transition profiles are needed to comprehensively quantify this effect.** Significantly different nonlinear dynamics (lower plots) and out-put spectra (upper plots) are observed in the three cases, highlighting the impact of dispersion variations on the output spectra and forming the scientific background for the concept. ...

B. Relevance of temperature transition region in the context of PSO-based optimization

In accordance with the Reviewer's comment, we would like to emphasize that a key strength of our approach lies in the fact that the PSO-based optimization operates independently of nonlinear pulse propagation simulations and relies solely on experimentally accessible parameters. As illustrated in the schematic of the optimization process (Fig. 3 of the main text), the central input for the optimization is the output spectrum, and precise knowledge of the temperature distribution within the LCF is experimentally not required. This renders the proposed optimization methodology practically insensitive to the exact shape of the temperature distribution in the transition regions between the heating elements and the surrounding air.

As this important point had not been clearly stated in the original manuscript, the main text has been revised accordingly.

Main text: ... In the experiment instead (yellow branch in Fig. 3), the fiber temperature is adjusted and the resulting output spectrum is measured by an automated data acquisition and control system, **leading to a framework that operates independently of nonlinear pulse propagation simulations and relies exclusively on experimentally accessible parameters. It is important to note that since the output spectrum serves as the sole input for the optimization, precise knowledge of the temperature distribution within the LCF is not experimentally required. This renders the proposed optimization methodology effectively insensitive to the exact shape of the temperature transition between the heating elements and the surrounding air.** Details on fiber preparation and optical setup can be found in the Methods section. ...

Conclusion: ... In particular, the experimental validation confirms the feasibility of implementing computationally optimized dispersion profiles in real fiber systems **using a framework that relies exclusively on experimentally accessible parameters**, paving the way for new advances in programmable light control in nonlinear photonic systems. ...

R1.3: Moreover, to make the manuscript more convincing, the authors should add at least one example of optimization (e.g., to achieve a flat spectrum over a specific interval) obtained both numerically and experimentally, and compare the spectra predicted by the numerical simulator with the measured ones.

We thank the Reviewer for the comment regarding a comparison between experiments and simulations which was also raised by Reviewer #3 in her/his first comment. While we agree that such comparisons are generally valuable for uncovering underlying physical mechanisms, we would kindly mention that the aim of this study is different: here, optimization is carried out independently of the specific LCF configuration, enabling optimized output spectra even when fiber parameters are not precisely known. This approach fundamentally differs from strategies that solely rely on nonlinear pulse propagation simulations for system optimization.

Given the presence of several experimental uncertainties that cannot be fully captured in simulations and the high dispersion sensitivity of the system studied, a direct match between experimental and simulated results is naturally limited. Although it would be possible to closely match both by fine-tuning simulation parameters, the underlying physical mechanisms are already well understood (soliton fission, dispersive wave formation, temporal interference), and practically no additional insights would be gained. Therefore, we refrain from a direct comparison and instead enhance our discussion of the potential experimental uncertainties that cannot be accounted for in the simulations.

We have performed additional nonlinear pulse propagation simulations (sech² pulse shape, 30 fs pulse duration, center wavelength 1560 nm, peak power 10 kW; 7.2 cm long fiber with 3.92 μm core diameter, TM₀₁ mode excitation, temperature profile as shown in Fig. 4c of the main text) that include temperature variations from the ideal distribution, showing that changes about $\pm 2^\circ$ indeed lead to spectrally different outputs (Fig. RL4). Such temperature variations are realistic for the experimental setup used, especially since the thermal coupling between the Peltier elements and the fiber may not be ideal or uniform across the entire surface of the respective Peltier element.

Fig. RL4: Nonlinear pulse propagation simulations showing generated output spectra for different configurations that include deviations from the ideal temperature distribution (see text for details). (a) Random temperature variations between -2°C and $+2^\circ\text{C}$ added to each temperature interval. (b) A constant temperature offset (values indicated in the legend) added to each temperature interval.

Furthermore, there are several critical parameters that cannot be readily incorporated into the simulations. For instance, the simulations assume a fully temperature-modulated 7.2 cm fiber (corresponding to 24 TECs), whereas the experiments include an unmodulated fiber segment at the input due to the presence of the optofluidic mount, as well as several centimeters of unheated fiber after the heated section. Additional uncertainties arise from simulation assumptions such as using an ideal sech^2 input pulse instead of the actual experimental pulse shape, unknown concrete power levels in the fiber, and potential cross-talk between Peltier elements.

To account for the Reviewer's comment, the text of the manuscript has been changed as follows:

... Note that the simulations are designed to highlight the system's capabilities rather than to precisely replicate experimental conditions. The simulations make unrestricted use of the parameter space, e.g. including precise temperature control and the absence of thermal disturbances. **This strategy ensures that simulations and experiments complement each other effectively, while experimental optimization is conducted independently of the specific LCF configuration, enabling optimized output spectra even when the fiber parameters are not precisely known. ...**

... In particular, the experimental validation confirms the feasibility of implementing computationally optimized dispersion profiles in real fiber systems, paving the way for new advances in programmable light control in nonlinear photonic systems. **From an experimental perspective, the optimization is performed independently of the specific LCF configuration, allowing optimized output spectra to be achieved even when the fiber parameters are not precisely known. ...**

Additionally, the section previously titled "Improvement of thermal control unit" has been renamed "**Improvement of experimental configuration**", and its content has been expanded to include a discussion of the experimental uncertainties that cannot be addressed in the simulations: ... Higher resolution could be achieved with alternative methods, like thermal printer heating elements, potentially extending applicability to effects requiring longer pulses, such as quasi-phase-matched harmonic generation. **Several critical experimental parameters cannot be fully incorporated into the simulations, making a direct comparison between experimental and simulated results challenging. These include temperature variations from the ideal distribution due to imperfect thermal coupling between the fiber and Peltier elements (c.f. Sec. 6 of the Supplementary Information), the presence of unmodulated fiber sections at the input and output in the experiment, and additional uncertainties such as assuming an idealized input pulse, unknown actual power levels in the fiber, and possible cross-talk between Peltier elements. ...**

The simulations of the configuration that include variations from the ideal temperature distribution including the corresponding discussion have been added to the Supplementary Information (Sec. 6).

We hope that this expanded discussion of experimental uncertainties demonstrates to the Reviewer that the PSO-based optimization approach can effectively optimize ultrafast

supercontinua, even when the fiber parameters and experimental conditions are not precisely known.

In my opinion, a revised version of the manuscript incorporating the above suggestions would fully meet the requirements for acceptance by Nature Communications.

We thank the Reviewer once again for the valuable comments and hope that our responses have addressed all concerns satisfactorily.

Response to comments of Reviewer #2

In the manuscript, authors presented a temperature sensitive liquid core fiber that combines with a heating array to control the mode dispersion segment by segment along the fiber, therefore termed as a programmable fiber with local dispersion control. Such a fiber opens the opportunity of computational optimization for ultrafast nonlinear interactions along the fiber, particularly the supercontinuum process that highly relies on the dispersion environment to tailor the overall spectral envelope. Indeed, authors have demonstrated by means of the particle swarm optimization the enhancement of power spectrum and the improvement of the spectral flatness. More detailed, they have also performed simulations that could confirm the experimental results, and have tested the spectrum stability over a long time period of few hours.

Such results are solid and could firmly support the novelty of the proposed scheme. I can't agree more that such a programmable fiber optic platform can be of broad interest, opening the way to the study of more complex nonlinear and soliton dynamics.

We appreciate the insightful comments and are committed to addressing each point raised. Below, we provide a detailed, point-by-point response to each comment and hope our answers address the Reviewer's concerns and further support our findings.

R2.1: Yet, it would be appreciated if authors could perform a more basic characterization on the specific nonlinear interaction on their programmable liquid core fiber. For example, they argued that the algorithm-led abrupt change of the zero dispersion points along the fiber is essential, which could excite more complex soliton dynamics such as the tunneling effects. But there lacks of a basic characterization or a showing example purely on the tunneling effects. From the simulation, it was still hard to identify the effect since the spectral evolution was very smoothing showing NO clear wavelength conversion via the tunnelling. So I would wonder what if taking a periodical changed ZDW as a standard characteristic to study the nonlinear interaction, which may provide more insights to the complex soliton dynamics.

We thank the Reviewer for the comments on soliton tunneling effects and periodic dispersion modulation, which we will address separately below.

A. Soliton tunneling effect

The Reviewer refers to a sentence in the Results section mentioning the soliton tunneling effect. To clarify, we would like to describing a scenario where a soliton propagates through three spatial regions: an initial anomalous dispersion region, followed by a normal dispersion domain where the Kerr effect no longer balances dispersion, and finally a section of

anomalous dispersion where the soliton reforms. In this context, the term “soliton reformation” more accurately describes the observed behavior, since “soliton tunneling” typically refers in the literature to spectral tunneling including energy conversion, which is not the case here. Please note that this topic is an area of ongoing research in our group and will be addressed in another publication.

To address the Reviewer’s comment, the main text has been revised as follows: ... Furthermore, significant reduction or disappearance of the AD region can trigger **soliton reforming after passing through a defined domain of normal dispersion, potentially accompanied by DW emission** [REFs]. ...

B. Periodic dispersion modulation

The second part of the Reviewer’s comment refers to periodic dispersion modulations as a means to study complex soliton dynamics. We agree that this is an excellent suggestion and note that our group has already explored this effect in another fiber system in the context of quasi-phase-matched higher-order DW generation [link]. Recently, we have also demonstrated this effect in LCFs [link], which makes us believe that quasi-phase-matched DW generation can also be observed using programmable fibers. As a result, we have recently initiated studies to investigate the dynamics of quasi-phase-matched DW generation in the programmable fiber platform.

In light of the Reviewer’s comment, we have changed the text as follows: ... The PSO-based optimization approach is not limited to SCG, but can also be applied to other nonlinear effects and waveguide systems [REF], such as higher harmonic generation or multimode fibers. **Since quasi-phase-matched higher-order DW generation has recently been demonstrated in LCFs with periodically modulated core diameters [link], the programmable fiber platform presented here could provide a valuable means to study this highly dispersion-sensitive phenomenon in a reconfigurable manner. Moreover, preliminary simulations of LCF configurations operating entirely in the normal dispersion regime show that the output spectra can be optimized for flatness in these cases, though further studies are needed to fully explore the capabilities and limitations.** Since the modulation mechanism is based on local changes in the waveguide dispersion, the method is also applicable in linear photonics, e.g. for the optimization of fiber-integrated spectral filters **or tunable dispersion compensation elements**. ...

R2.2: Another minor comment is on the spectral flattening example. In the presented result, it seems the dispersive waves are still dominating having the strongest spectral intensity. Is it possible to program the fiber in the normal dispersion regime where spectral broadening usually shows decent flatness compared with that in the anomalous dispersion in the soliton regime? The current solution seems only working for sideband spectrum around the pumping wave (and should not be overlapped with the dispersive wave)?

We thank the Reviewer for the comment concerning spectral flattening and the impact of the dispersive wave. In response, we first address the Reviewer’s suggestion to simulate the concept’s capability for optimizing spectral output in the case of a normal dispersive fiber (Sec. A), followed by a discussion of the impact of the dispersive wave (Sec. B).

A. Demonstration of the PSO-based optimization concept in the normal dispersion domain

One comment of the Reviewer addresses whether the presented PSO-based optimization concept can also be applied to the optimization of output spectra in the case of normal dispersion (ND). We agree that this is a highly interesting suggestion and, in principle, it is entirely possible and could be explored in future work. Generally, optimizing supercontinua generated in the normal dispersion regime is feasible using the same approach, although, based on our experience, the temperature sensitivity of nonlinear frequency conversion processes in this domain is lower compared to soliton-based effects.

To demonstrate in principle that PSO-based optimization can be applied to ND scenarios (Fig. RL5), several test optimizations were performed using the TM_{01} mode of a CS_2 -LCF with a core diameter of $3.6\ \mu\text{m}$, which maintains ND throughout the full temperature range from $15\ ^\circ\text{C}$ to $45\ ^\circ\text{C}$ (sech² pulse shape, pulse duration 30fs, center wavelength 1560nm, peak power 10kW).

Fig. RL5: Two simulation examples of optimized output spectra using the TM_{01} mode of a CS_2 -LCF with a core diameter of $3.6\ \mu\text{m}$, where the dispersion remains all-normal across the relevant spectral range for all temperatures considered (dashed line: room temperature, solid line: optimized spectrum). The light blue background indicates the predefined optimization intervals.

The results clearly demonstrate that even in the case of entirely normal dispersion, the output spectra can be optimized for flatness. While these simulations confirm the applicability of the concept to ND scenarios, further studies are necessary to fully assess its potential, which will be the focus of future work. Therefore, we have chosen not to include these simulation results in the present manuscript and instead provide an outlook on future experiments in the main text.

We also emphasize that the concept of programmable fiber is not limited to nonlinear photonics as the ultrafast supercontinuum generation explored here is just one illustrative example. Programmable fibers likely have a much wider range of applications and may be used in various fields, including linear optics, where they could serve as tunable dispersion filters and tunable dispersion compensation elements.

To account for the Reviewer's comment, the text has been changed as follows: ... The PSO-based optimization approach is not limited to SCG, but can also be applied to other nonlinear effects and waveguide systems [REF], such as higher harmonic generation or multimode fibers. **Since quasi-phase-matched higher-order DW generation has recently been demonstrated in LCFs with periodically modulated core diameters [link], the programmable fiber platform presented here could provide a valuable means to study**

this highly dispersion-sensitive phenomenon in a reconfigurable manner. Moreover, preliminary simulations of LCF configurations operating entirely in the normal dispersion regime show that the output spectra can be optimized for flatness in these cases, though further studies are needed to fully explore the capabilities and limitations. Since the modulation mechanism is based on local changes in the waveguide dispersion, the method is also applicable in linear photonics, e.g. for the optimization of fiber-integrated spectral filters or tunable dispersion compensation elements. ...

B. Suppression of dispersive wave through PSO-optimization

We would like to point out that the DW consistently appears as a strong feature in the output spectra, while the optimization process significantly reduces its amplitude. For example, in Fig. 5(a) of the manuscript (Fig. RL6), a pronounced DW is present at around 2.45 μm at room temperature. After optimization, this DW is flattened and no longer visible in the spectral distribution. Note that increasing the temperature generally causes a blue shift of the long-wavelength DW, which is not apparent in the flattened spectra after the optimization.

Fig. RL6 (excerpt from manuscript, Fig. 5(a)): Simulation results of PSO targeting flat spectral output at high power levels according to Eq. 5. (a) Example ($\Delta\lambda = 400$ nm, $\lambda_t = 2100$ nm) showing the optimized output spectrum (solid line) compared to the unoptimized spectrum at room temperature (dashed line). The light red region indicates the predefined optimization interval.

To address the Reviewer's comment, the spectral interval shown in Fig. 5(a) has been expanded and the main text has been updated accordingly. ... Figure 5a shows an example with an exceptionally flat region extending beyond the target range ($F_{\text{opt}} = 0.036$, solid line), which is a significant improvement over the spectrum of the unmodulated fiber at room temperature ($F_{\text{RT}} = 0.987$, dashed line). **Notably, the DW at 2.45 μm in the room temperature case, which consistently appears as a strong feature in the output spectra, is flattened and no longer visible after optimization.** Note that similar to the previous case O1, the non-intuitive temperature distribution, (Fig. 5c) and the associated GVD result in highly complex soliton dynamics (Fig. 5b), driven by the nonlinear frequency conversion processes mentioned earlier. ...

To conclude, I would be supportive to accept this work that are certainly of high novelty and of broad interest, if provided with in-depth and demonstrated understandings of the "non-intuitive" designs.

We thank the Reviewer for the supportive feedback and constructive suggestions, and hope that our revisions and clarifications have addressed the Reviewer's concerns.

Response to comments of Reviewer #3

The manuscript by Hofmann et al. presents a novel approach for tailoring the shape of a nonlinearly generated fiber supercontinuum spectrum through longitudinal shaping of the core refractive index of the (liquid-filled) fiber by temperature control. A complex longitudinal dispersion map can in this way be realized, and numerically optimized according to specified criteria. The authors demonstrate optimization of either power or spectral flatness in specific wavelength bands, both numerically and experimentally. Some directions for further research are briefly discussed.

The application of a complex spatial temperature profile, and its optimization using contemporary algorithms (swarm optimization) is to my knowledge novel, and could hold significant potential beyond the particular application demonstrated in the paper. The numerical and experimental procedures as described appear sound to me, and the authors make reasonable claims based on the data obtained. I believe that appropriate detail about the simulations and experiments is given for other researchers to attempt a reproduction of the results, but some of my subsequent comments should be taken into account here.

We are grateful to the Reviewer for the careful evaluation and favorable opinion of our manuscript. We value the constructive feedback provided and will address every point in detail. In the following, we present a comprehensive, point-by-point reply to each comment and hope that our responses resolve the Reviewer's concerns and further validate our results.

My main concerns about the work and results as presented are the following:

R3.1: Comparison between theory and experiment: The authors state that the 'simulations are designed to highlight the system's capabilities rather than to precisely replicate experimental conditions.' This is all very well, and I know from direct personal experience how difficult it can be to reconcile simulations and measurements. However, given that the simulations appear to resemble the experimental layout quite accurately, I am a bit concerned about the total lack of correspondence between simulations and experiments. As a particular example, the authors optimize a flatness factor of 0.036 (lower is better) in the spectral range 1900-2300 nm based on simulations. However, their best experimental result is a flatness factor of 0.26, and in a completely different spectral window, namely 1300-1500 nm. Surely, if the authors were anywhere close to reproducing the simulated result, they would have shown it. This raises the question if the simulations can be trusted at all, and at the very least, a much more extended discussion of their accuracy, and the possible reasons for discrepancies is needed. In particular, I think there should be some direct comparisons between particular experimental spectra, and corresponding best efforts to reproduce them in simulations.

We thank the Reviewer for the comment regarding a comparison between experiments and simulations which was also raised by Reviewer #3 in her/his first comment. While we agree that such comparisons are generally valuable for uncovering underlying physical mechanisms, we would kindly mention that the aim of this study is different: here, optimization is carried out independently of the specific LCF configuration, enabling optimized output spectra even when fiber parameters are not precisely known. This approach fundamentally differs from strategies that solely rely on nonlinear pulse propagation simulations for system optimization.

Given the presence of several experimental uncertainties that cannot be fully captured in simulations and the high dispersion sensitivity of the system studied, a direct match between experimental and simulated results is naturally limited. Although it would be possible to closely match both by fine-tuning simulation parameters, the underlying physical mechanisms are already well understood (soliton fission, dispersive wave formation, temporal interference), and practically no additional insights would be gained. Therefore, we refrain from a direct comparison and instead enhance our discussion of the potential experimental uncertainties that cannot be accounted for in the simulations.

We have performed additional nonlinear pulse propagation simulations (sech² pulse shape, 30 fs pulse duration, center wavelength 1560 nm, peak power 10 kW; 7.2 cm long fiber with 3.92 μm core diameter, TM₀₁ mode excitation, temperature profile as shown in Fig. 4c of the main text) that include temperature variations from the ideal distribution, showing that changes about $\pm 2^\circ$ indeed lead to spectrally different outputs (Fig. RL7). Such temperature variations are realistic for the experimental setup used, especially since the thermal coupling between the Peltier elements and the fiber may not be ideal or uniform across the entire surface of the respective Peltier element.

Fig. RL7: Nonlinear pulse propagation simulations showing generated output spectra for different configurations that include deviations from the ideal temperature distribution (see text for details). (a) Random temperature variations between -2°C and $+2^\circ\text{C}$ added to each temperature interval. (b) A constant temperature offset (values indicated in the legend) added to each temperature interval.

Furthermore, there are several critical parameters that cannot be readily incorporated into the simulations. For instance, the simulations assume a fully temperature-modulated 7.2 cm fiber (corresponding to 24 TECs), whereas the experiments include an unmodulated fiber segment at the input due to the presence of the optofluidic mount, as well as several

centimeters of unheated fiber after the heated section. Additional uncertainties arise from simulation assumptions such as using an ideal sech^2 input pulse instead of the actual experimental pulse shape, unknown concrete power levels in the fiber, and potential cross-talk between Peltier elements.

To account for the Reviewer's comment, the text of the manuscript has been changed as follows:

... Note that the simulations are designed to highlight the system's capabilities rather than to precisely replicate experimental conditions. The simulations make unrestricted use of the parameter space, e.g. including precise temperature control and the absence of thermal disturbances. **This strategy ensures that simulations and experiments complement each other effectively, while experimental optimization is conducted independently of the specific LCF configuration, enabling optimized output spectra even when the fiber parameters are not precisely known. ...**

... In particular, the experimental validation confirms the feasibility of implementing computationally optimized dispersion profiles in real fiber systems, paving the way for new advances in programmable light control in nonlinear photonic systems. **From an experimental perspective, the optimization is performed independently of the specific LCF configuration, allowing optimized output spectra to be achieved even when the fiber parameters are not precisely known. ...**

Additionally, the section previously titled "Improvement of thermal control unit" has been renamed "**Improvement of experimental configuration**", and its content has been expanded to include a discussion of the experimental uncertainties that cannot be addressed in the simulations: ... Higher resolution could be achieved with alternative methods, like thermal printer heating elements, potentially extending applicability to effects requiring longer pulses, such as quasi-phase-matched harmonic generation. **Several critical experimental parameters cannot be fully incorporated into the simulations, making a direct comparison between experimental and simulated results challenging. These include temperature variations from the ideal distribution due to imperfect thermal coupling between the fiber and Peltier elements (c.f. Sec. 6 of the Supplementary Information), the presence of unmodulated fiber sections at the input and output in the experiment, and additional uncertainties such as assuming an idealized input pulse, unknown actual power levels in the fiber, and possible cross-talk between Peltier elements. ...**

The simulations of the configuration that include variations from the ideal temperature distribution including the corresponding discussion have been added to the Supplementary Information (Sec. 6)

We hope that this expanded discussion of experimental uncertainties demonstrates to the Reviewer that the PSO-based optimization approach can effectively optimize the ultrafast supercontinua, even when the fiber parameters and experimental conditions are not precisely known.

Please note that the value of the flatness factor given in the main text should match the value reported in the table. The correct value for the experimental flatness factor is 0.61, which has been updated in the revised manuscript.

R3.2: Reproducibility: The authors state that results for their key metrics show some day-to-day variation, due to drifts and inaccuracies in the optical setup. This should be better quantified. For instance, if one of their eta-parameters is optimized from scratch on different days, what is the magnitude of deviations? How many of the significant figures given in, e.g. Table 1 can we actually trust?

We thank the Reviewer for the comment regarding reproducibility. After reviewing the manuscript in light of this comment, we realize that we did not sufficiently address this point in the manuscript, and the observed day-to-day changes could be misinterpreted as degradation or alteration of the core liquid due to laser exposure.

We would like to note that day-to-day variations are primarily caused by thermal drift of the coupling optics over time, which alters the coupling conditions and consequently the supercontinuum spectral shape. This is not due to degradation of the core liquid or any other system component, and careful realignment can nearly restore the initial output spectrum. To demonstrate the stability of both the experimental system itself and the liquid under laser exposure, we have recently published a study showing stable supercontinuum generation in a CS₂-LCF operating in a higher-order mode over many hours [link]. Additionally, we have published a study on the stability of CS₂-based LCFs when exposed to very high average laser powers (>1 W [link]), showing no degradation over the entire duration of the measurements (several days).

We believe that the observed day-to-day fluctuations can be largely eliminated by integrating the LCF into fiber circuitry via splicing, as we recently demonstrated in experiments related to Brillouin-Mandelstam scattering [link]. Therefore we believe that additional experiments to quantify deviations in the reported benchmark figures are not useful, as these fluctuations are not intrinsic to the PCF platform but arise from limitations of the coupling optics, which can be addressed through photonic integration. Therefore, the reported benchmark parameters are reliable and are expected to improve further with the implementation of integrated fiber coupling.

To account for the Reviewer's comment, the main text of the manuscript has been changed as follows: ... It should be noted that room temperature spectra can vary between different optimization runs, resulting in different optimized spectra despite identical optimization parameters and objectives. This variation is mainly due to different coupling conditions as the measurements were performed on different days. **It is important to note that these variations are not due to degradation of the LCF or the core liquid, as recent studies have shown stable supercontinuum generation in a CS₂-LCF operating in a higher-order mode over many hours [link], as well as stable operation over several days under exposure to very high average laser powers (>1 W [link]).** The variation of the setup over the course of a measurement is minimal, ensured by the advanced setup design and in particular by the piezo-stabilized coupling [46]. This stability has been confirmed by control spectra recorded at room temperature at the end of each iteration by switching off the Peltier elements, which show sufficient stability over the duration of the

experiments (see Supplementary Information Sec. 4), ensuring reliable optimization throughout the measurement period. ...

R3.3: Flexibility: The authors discuss that the simulations assume step-like temperature changes, whereas in reality this is not possible. Can one derive fundamental limits to the steepness of temperature gradients based on the physical properties of the medium, and reasonable assumptions about the feature sizes of heaters, fiber cores, etc.?

We thank the Reviewer for highlighting the issue of step-like temperature changes assumed in our simulations, which was also mentioned by Reviewer #1 in her/his first comment (R1.1). To address this, we first outline the simulation methodology (Sec. A), then analyze the steepness of the temperature gradient in the context of the LCF used in this study (Sec. B), and finally discuss the influence of geometric and material parameters on the resulting temperature distribution (Sec. C).

A. Description of simulations: To reveal the internal temperature distribution within the LCF sample, we conducted numerical simulations of the steady-state temperature profile in a 2D silica glass block using a Python-based finite-difference scheme. The model discretizes the Laplace equation on a rectangular grid, with periodic boundary conditions applied along the fiber axis (z -direction). The bottom boundary ($x = 0$) is maintained at a spatially periodic temperature profile ($T_0 = 20^\circ\text{C}$, $T_1 = 80^\circ\text{C}$) to simulate external heating, implementing Dirichlet boundary conditions. The top boundary ($x = 125 \mu\text{m}$, matching the experimental LCF diameter) employs a Robin (mixed) boundary condition to account for convective heat transfer to the surrounding air ($T_{\text{air}} = 20^\circ\text{C}$). The numerical solution is obtained using a custom Python code by assembling and solving the resulting sparse linear system for the interior points, with all boundary conditions directly incorporated. This approach provides a physically realistic model of the thermal behavior of an infinitely extended silica glass block in contact with air on one side, closely mirroring experimental conditions and incorporating the thermal conductivity of silica ($k_{\text{silica}} = 1.31 \text{ W}/(\text{m}\cdot\text{K})$ [link]) as well as the heat transfer coefficient at the silica–air interface ($h_{\text{heat}} = 10 \text{ W}/(\text{m}^2\cdot\text{K})$ [link]).

B. Temperature distribution in the LCF used in this study

Fig. RL8: Simulated temperature distribution within a silica glass block, with periodic Dirichlet boundary conditions applied at the bottom ($x = 0$) and Robin boundary conditions at the top surface ($x = 125 \mu\text{m}$). To reflect the experimental configuration, the pitch Λ was set to 5.6 mm, matching twice the length of one Peltier element (2.8 mm). Further simulation details are described in the text. (a) Temperature map in the xz -plane at $y = 0$. (b) Longitudinal temperature profile at half the size of the

glass block ($x = 62.5 \mu\text{m}$, indicated by horizontal dashed green lines in (a)). The horizontal dashed gray lines indicate the maximum and minimum temperatures considered (20°C and 80°C).

The resulting 2D temperature distribution using a periodicity Λ of 5.6 mm (shown in Fig. RL8(a)) demonstrates that the transverse temperature variation along the x -axis is relatively minor, primarily due to the small outer diameter of the fiber ($125 \mu\text{m}$). This effect becomes even more apparent when examining the longitudinal temperature profile at the midpoint of the block ($x = 62.5 \mu\text{m}$, Fig. RL8(b)), which shows only slight deviations from the ideal step-like profile at the curve edges.

C. Temperature distribution in different LCF configurations

Impact of materials: To assess the impact of material parameters on the temperature distribution, it is useful to refer to the simulations described in Sec. B, which are fundamentally based on solving the Laplace equation to determine the steady-state temperature profile. These simulations use periodic boundary conditions along the z -axis to represent periodic heating, with Dirichlet and Robin boundary conditions applied at the bottom and top interfaces to simulate heating from below and exposure to an open environment above. The only direct influence of material properties in these simulations arises in the Robin boundary condition at the top, which models convective heat dissipation into air using the thermal conductivity of silica and the heat transfer coefficient at the silica–air interface. As shown in Fig. RL7 (a), the temperature distribution along the x -direction remains largely uniform, with only minimal deviations near the uppermost region of the glass ($x \leq 125 \mu\text{m}$), indicating that the impact of material properties is generally minor.

Impact of geometric features: Figure RL9 presents simulated longitudinal temperature distributions at the center of the glass block ($x = 62.5 \mu\text{m}$), corresponding to the location of the fiber core, for different geometric configurations. As shown in Fig. RL9 (a), small block extensions (e.g., $80 \mu\text{m}$, $125 \mu\text{m}$) result in temperature profiles that closely approximate a step-like function, whereas larger extensions lead to smoother profiles due to increased thermal diffusion.

Fig. RL9: Simulated longitudinal temperature distributions at half the extension (along the x -axis, $x = 62.5 \mu\text{m}$) of the glass block, corresponding to the fiber core position (see Sec. A for simulation details). (a) Temperature profiles for different glass block extensions (indicated in the legend in micrometers, $\Lambda = 5.6 \mu\text{m}$). (b) Temperature profiles for different periodicities (pitch values shown in the legend in

micrometers). The x-axis is normalized to the respective pitch to allow direct comparison of the profiles.

Figure RL9(b) shows temperature distributions for different periodicities (pitch values), indicating that for comparably large pitches - such as the experimental value of $\Lambda = 5.6 \mu\text{m}$ - the step-like character of the temperature distribution is preserved, and cross-talk between regions of different temperature remains minimal. Noticeable smoothing only appears for much smaller pitch values.

These findings confirm that, under the experimental parameters used ($\Lambda = 5.6 \mu\text{m}$, extension $125 \mu\text{m}$), the temperature distribution remains sharply step-like and cross-talk between heating elements is small, thereby validating the experimental design.

In response to the Reviewer's comment, we have added these sections including the figures to the Supplementary Information (new Sec. 1) and revised the main text accordingly:

... Due to the arrangement of the heating elements used in the experiments (24 Peltier elements, approx. length 2.8 mm each), the GVD changes discretely, resulting in a stepwise variation of the ZDWs, indicated by dashed white lines in Fig. 2. **It should be noted that additional 2D finite-element simulations of the temperature distribution in a comparable silica glass block for various LCF geometries (see Sec. 1 of the Supplementary Information) reveal only minor deviations from a step-like longitudinal temperature profile when using the experimental fiber parameters, supporting the validity of the step-like temperature distribution assumption. Example nonlinear pulse propagation simulations revealed that the generated output spectra are largely insensitive to the specific shape of the temperature transition at the heating element-air interface, although future studies with full statistical analysis of different transition profiles are needed to comprehensively quantify this effect.** Significantly different nonlinear dynamics (lower plots) and out-put spectra (upper plots) are observed in the three cases, highlighting the impact of dispersion variations on the output spectra and forming the scientific background for the concept. ...

R3.4: Response time: The authors claim reconfigurability as a major achievement of the work, but it seems to me that the reconfiguration time is quite slow, ranging from at least 7 seconds if the desired temperature profile is known, to several hours if a reoptimization is needed. This will be a significant restriction of potential use cases. The manuscript contains some discussion of potential improvements, but is it possible to put some numbers on that, e.g. a lower bound on the reconfiguration times obtainable?

We thank the Reviewer for the comment regarding the device's response time, which, as noted, has already been addressed in the manuscript. In the following, we discuss different possibilities including relevant numbers and finally highlighted the changes to the main text:

Light-induced temperature modulation: As reported by Y. Wan *et al.* [link], the time scale for light-induced temperature changes is on the order of 1 s, which is considerably faster than the measurement time of 7 s observed in our current experimental setup, mainly resulting from the thermal inertia of the Peltier elements.

Improved heating elements: An alternative and practically relevant approach to enhance time response is the use of thermal print heads, which are widely used in applications such as barcode and receipt printing and offer response times in the range of 1–5 ms [link]. This considerable improvement over Peltier elements is due to their much lower thermal mass and the direct resistive heating mechanism, rather than the thermoelectric process of Peltier devices. Metallic wires also provide rapid heating due to their low thermal mass. Additionally, microfabricated resistive microheater arrays, such as metal or ceramic thin films, integrated directly onto the fiber surface can further decrease both the thermal mass and the distance between the heater and the waveguide core, allowing for rapid and highly localized temperature modulation, potentially on millisecond time scales (e.g. a Si-based thin-film microheater with a thermal rise time under 20 ms was demonstrated in Ref. [link]).

To account for the Reviewer's comment, we have included these points into the main text as follows: ... A challenge of the present experimental setup is the relatively long measurement time, primarily caused by the slow transition to thermodynamic equilibrium of 7 seconds on average. **The slow time response is primarily due to the properties of the Peltier elements used, which have relatively large thermal mass and inertia. An alternative solution is to employ thermal print heads, commonly used in applications such as barcode and receipt printing, which offer response times in the range of 1–5 ms [link]. This improvement results from their lower thermal mass and direct resistive heating mechanism, as opposed to the thermoelectric operation of Peltier devices. Metallic wires similarly enable rapid heating because of their low thermal mass. Furthermore, integrating microfabricated resistive microheater arrays - such as metal or ceramic thin films - directly onto the fiber surface could enable fast and highly localized temperature modulation, potentially on millisecond time scales (e.g. a Si-based thin-film microheater with a thermal rise time under 20 ms was demonstrated in Ref. [link]).**

In addition, light-to-heat approaches to generate temperature distributions may also help to shorten relaxation times. Recent studies have shown that coating LCF with carbon nanotubes allows modulation of the supercontinuum radiation by external light irradiation on timescales of a few seconds (**e.g., 1s response time was demonstrated in Ref. [link]**). In principle, this approach could be further accelerated by introducing light-absorbing materials (e.g. graphene) directly into the liquid core.

In this study, the spatial resolution of the temperature distribution is defined by the width of the Peltier elements (2.8 mm), which is sufficient for the current experimental configuration, i.e., the specific combination of LCF parameters and pulse properties. If required, higher resolution could be achieved with alternative methods, like thermal printer heating elements, potentially extending applicability to effects requiring longer pulses, such as quasi-phase-matched harmonic generation. ...

R3.5: Generality: The proposed approach could in principle find many applications, and open a new sub-field of nonlinear fiber optics. In this sense, publication in a journal like Nature Comm. would seem appropriate. However, in order to optimize the effect of temperature variations the authors have used a particular (albeit common) nonlinear medium, CS₂, known for its high temperature coefficient. They have further optimized the temperature sensitivity through the use of a particular fiber mode, namely the TM₀₁. Now, the use of CS₂ (or any other particular material) invariably implies restrictions in terms of

transmission windows, dispersion profiles etc. It is also noteworthy that CS₂ is rather poisonous. The use of the TM₀₁ mode implies issues with in/outcoupling, in particular if a broadband continuum needs to be brought back to a gaussian-like profile. So I think that at least a semi-quantitative discussion of other media is needed to judge the actual impact and generality of the idea. What level of tunability can be expected, e.g. with commonly used infiltration liquids, what can be achieved in the fundamental mode, or other mode types, etc.

We thank the Reviewer for the comment concerning the use of LCFs with CS₂ and the consideration of other potential material systems for fiber-based temperature modulation. In the following, we first address the properties of CS₂ in the context of ultrafast nonlinear frequency conversion and then discuss alternative material options.

A. Capabilities and limitations of CS₂ in the context of ultrafast photonics in liquid-core fibers

Toxicity: As correctly noted by the Reviewer, CS₂ is indeed toxic when present in large quantities. However, it is important to emphasize that the amount used in the LCF is extremely small—on the order of nanoliters. For example, a 10 cm long cylindrical liquid column with a diameter of 3.92 μm contains only 1.2 nL of liquid, which is well below typical safety limits. Consequently, integrating a CS₂-based LCF into a spliced fiber environment (as demonstrated by our group in Ref. [link]) effectively mitigates concerns regarding toxicity.

Spectroscopic properties: As demonstrated in our previous work on the spectroscopic properties of CS₂ and other inorganic materials [link], CS₂ exhibits broad transmission windows extending well into the mid-IR, surpassing the transmission range of silica. Figure RL10 illustrates this with a representative excerpt from the mentioned publication showing the attenuation coefficient spectrum of CS₂ from 1.5 μm to 20 μm.

Fig. RL10: Attenuation spectrum of CS₂ in the near- and mid-infrared regions (data from our work [link]). The horizontal gray dotted line in each plot marks an attenuation level of 1 dB/cm. Gray-shaded areas indicate regions where no data was recorded due to excessively high attenuation (top: linear scale, bottom: logarithmic scale).

CS₂ is also well suited for applications in the visible spectral domain, as it is effectively transparent throughout the visible and near-infrared regions. This has been demonstrated, for example, in the work of R. Ganeev *et al.* [link], which reports that CS₂ exhibits low absorption for wavelengths longer than 400 nm across the visible spectrum (Fig. RL11).

[REDACTED]

Fig. RL11: Optical absorption spectrum of CS₂ in the visible spectral domain (taken from [link]). The inset shows a schematic diagram of the experimental setup.

As a result, our experience shows that when short fibers (<20 cm) are used, CS₂/silica LCFs offer a spectral operation range from 400 nm to 4 μm, which exceeds the range available in conventional silica glass fibers.

To account for the Reviewer's comment, the main text of the manuscript has been amended as follows: ... The pLCFs used in this study consist of a 13 cm long, in-house fabricated, fiber-like silica capillary with an inner diameter of 3.92 μm, which is filled with CS₂ by capillary action within minutes. **Note that although CS₂ is toxic in large quantities, the volume used in the LCF is extremely small (e.g., only 1.2 nL for the 10 cm LCF discussed here), rendering toxicity concerns negligible when the fiber is integrated into a spliced environment [link]. The core diameter was chosen to ensure that the radially polarized TM₀₁-mode exhibits two ZDWs at room temperature to position the central pulse wavelength $\lambda_p = 1560$ nm in the AD region, thus allowing soliton-based effects (c.f. [REF] for details). **Note that when considering short fibers (<20 cm), the high transparency of CS₂ enables spectral operation from 400 nm to approximately 4 μm [link], [link]. ...****

B. Ultrafast nonlinear frequency conversion using higher-order modes in LCF

The experiments presented in this work utilize a higher-order mode, which exhibits significant temperature-dependent dispersion. This choice is crucial, as the fundamental mode of CS₂ LCFs displays far less pronounced temperature sensitivity, as for example shown in our previous work on dispersive wave tuning [link].

As the Reviewer correctly pointed out, some applications may require a Gaussian-like fundamental mode, where the use of higher-order modes is critical. In this context, we would like to mention our recent research on mode conversion using 3D nanoprinted, fiber-integrated nanostructures. As demonstrated in [link] and [link], phase-only holograms or metasurfaces can be directly nanoprinted onto the fiber end face to generate complex beam shapes. This approach principally enables the conversion of higher-order modes into fundamental modes, and is currently being pursued as a research project in our group. We

therefore consider fiber-integrated mode conversion to be a promising strategy for applications that require access to the fundamental mode.

To account for the Reviewer's comment, the main text has been changed as follows: ... Although this method allows for dispersion modulation, its flexibility and tuning range are inherently limited, and multimode operation may not be suitable for certain applications. In addition, mechanical adjustments can present challenges such as potential fiber damage and reduced dispersion control accuracy. **Note that the use of a higher-order mode in this work does not limit the application range of the LCF platform, as recent advances in 3D nanoprinting enable the fabrication of efficient on-fiber mode converters - such as phase-only holograms [link] or metasurfaces [link] - that can readily convert higher-order modes to fundamental modes. ...**

C. Other potential materials and fiber structures

A critical parameter in the programmable fiber concept is the use of a material with a very high thermo-optic coefficient (TOC), enabling modal dispersion to be manipulated via tailored temperature patterns. To provide a comprehensive overview of suitable materials, Tab. RL1 summarizes TOCs for a range of photonic materials relevant to fiber optics, including liquids, chalcogenide glasses, and semiconductors. Notably, liquid CS₂ exhibits a very large negative TOC ($-8 \times 10^{-4} \text{ K}^{-1}$) [link], making it extremely responsive to temperature changes, while other liquids such as benzene [link] and TCE [link] also show strong negative TOC values.

Table RL1: Comparison of TOCs for various photonic materials relevant to fiber optics, including liquids, chalcogenide glasses, and semiconductors. The table summarizes the magnitude and sign of the TOCs, the measurement wavelengths, and the relevant literature sources.

Material (Composition)	class	dn/dT (K ⁻¹)	Transparency range
CS ₂	liquid	$\sim -8 \times 10^{-4}$ [link]	VIS to NIR [link]
C ₂ Cl ₄	liquid	$\sim -6 \times 10^{-4}$ [link]	VIS to NIR [link]
CCl ₄	liquid	$\sim -3.9 \times 10^{-4}$ [link]	VIS to NIR [link]
Toluene	liquid	$\sim -5.5 \times 10^{-4}$ [link]	Up to 1.6 μm [link][link][link][link]
As ₂ S ₃	chalcogenide glass	$\sim 9 \times 10^{-6}$ [link]	1-7 μm [link]
As ₂ Se ₃	chalcogenide glass	$\sim 3.2 \times 10^{-5}$ [link]	2-13 μm [link]
Ge ₃₃ As ₁₂ Se ₅₅	chalcogenide glass	$\sim 7.6 \times 10^{-5}$ [link][link] [link]	1-15 μm [link][link]
Si	semiconductor	$\sim 1.9 \times 10^{-4}$ [link][link]	1 - 8 μm [link][link][link][link][link]
Ge	semiconductor	$\sim 5 \times 10^{-4}$ [link]	1.5-15 μm [link][link][link][link]
GaAs	semiconductor	$\sim 2.3 \times 10^{-4}$ [link] [link] [link] [link]	1-16 μm [link] [link]

For comparison, silicon has a positive TOC of $+1.9 \times 10^{-4} \text{ K}^{-1}$ at 1.55 μm [link], which is significant for solid-state materials and similar in order of magnitude to CS₂. Germanium offers an even higher TOC than Si ($+5 \times 10^{-4} \text{ K}^{-1}$), while its high absorption at near-IR wavelengths excludes it from waveguiding applications in this range.

The chalcogenide glass Ge₃₃As₁₂Se₅₅ shows a positive TOC of $+7.6 \times 10^{-5} \text{ K}^{-1}$ in the mid-infrared ($\approx 4\text{--}10 \mu\text{m}$) [link], making it useful for mid-IR applications, though with lower temperature sensitivity than CS₂. Note that As₂S₂, already used in hybrid chalcogenide-silica

fibers [link], has a much lower TOC [link], making it less suitable for temperature-sensitive applications. This comparison clearly justifies the choice of CS₂ as the core material, as it offers the highest TOC among feasible high-quality step-index fiber materials.

Preliminary simulations investigating the temperature dependence of the group velocity dispersion (GVD) in liquid, semiconductor, and chalcogenide glass cylindrical core fibers with silica cladding reveal the highest temperature sensitivity for CS₂/silica fibers in higher-order modes—a finding that is currently under further investigation. Other fiber types exhibit substantially lower temperature dependence due to the complex interplay of refractive index, TOC, and waveguiding properties. At present, we consider the CS₂–silica step-index fiber to provide the highest temperature susceptibility, although ongoing research is aimed at more fully characterizing the temperature-dependent properties of alternative core materials. We are currently conducting additional simulation-based studies to address this question in detail.

We would also like to point out that LCFs can be realized with microstructured claddings to tailor the GVD profile. This was recently demonstrated by supercontinuum generation in a liquid-core microstructured optical fiber using CS₂ as the core material [link], which enabled a dispersion landscape with a zero-dispersion wavelength for the fundamental mode near the telecom range—unlike conventional capillary-type fibers, which have less favorable dispersion properties for soliton fission. Further research is needed to determine whether tailored external temperature patterns can be effectively transferred to the fiber core in the presence of a holey cladding, and whether the temperature sensitivity of the guided modes—ideally the fundamental mode—can be further enhanced by the use of a microstructured cladding.

To address the Reviewer's comment, we have added this table including the corresponding text to the Supplementary Information as Sec. 7 and expanded the main text accordingly. ... Another relevant type of waveguide are hybrid fibers containing unconventional materials [REF], which can also be used for temperature modulation in case of suitable core materials. **Examples of such hybrid waveguides include silicon or chalcogenide core fibers [link], [link], [link], which in principle offer very high thermo-optic coefficients along with strong nonlinearities, making them potentially relevant for programmable fibers and temperature-sensitive waveguide applications. A detailed comparison of the properties of materials with high thermo-optic coefficients is provided in Sec. 7 of the Supplementary Information. Note that it has recently been shown that LCFs with microstructured claddings can be fabricated to potentially enhance temperature sensitivity [link], though further investigation is needed to determine whether tailored external temperature patterns can be efficiently transferred to the fiber core and if the temperature sensitivity of the guided modes—ideally the fundamental mode—can be further improved by using a microstructured cladding. ...**

Overall, my impression of the paper is fairly positive, but I think that a proper discussion of the above points is needed to allow both readers and editors to judge the reliability and potential impact of the work.

We appreciate the Reviewer's positive assessment and thoughtful comments, and hope that our comprehensive responses and expanded discussion have fully addressed all of the Reviewer's concerns.

Response to comments of Reviewer #1

In their response, the authors show through numerical simulations how a periodic temperature variation can be achieved under boundary conditions compatible with their experiment, in which the temperature inside a liquid core fiber is controlled by Peltier cells arranged periodically. Moreover, the simulations included in the “supplementary information” demonstrate that periodic step-like temperature profiles are also feasible. In their heat diffusion simulations, the authors considered a 2D silica block rather than a silica capillary filled with carbon disulfide, but in my opinion, the results presented are still convincing, as they confirm the periodic distribution of temperature and how it is influenced by the capillary diameter (Fig. S2(a)) and the spacing between the Peltier cells (Fig. S2(b)).

I believe the revised version of the manuscript can be accepted by Nature Communications.

We thank the Reviewer for the thorough evaluation of our revisions and manuscript and greatly appreciate the recommendation to accept the current version.

Response to comments of Reviewer #2

I appreciate author's effort to make such a comprehensive and detailed reply to all the comments including mines, and I am satisfied with their revisions on the manuscript.

We thank the Reviewer for the detailed reading of the revisions and manuscript, as well as for the constructive feedback provided throughout the review process.

R2.1: A remaining question is that how to further extend the spectral range of optimization particularly when with a purpose of spectrum flatening? With presented results, the optimization range seems still quite narrowed if compared with the full range of the supercontinuum. It would be better if authors could give a brief discussion on this question in the manuscript (e.g. in the conclusion part).

We thank the Reviewer for the comment regarding the spectral extent of the optimization in the context of achieving a flat output spectrum. We would like to note that tuning the group velocity dispersion of a fiber is a powerful tool, yet inherently limited by the fundamental physical constraints of phase matching and soliton dynamics, which cannot be arbitrarily altered through local dispersion control. Furthermore, dispersion tuning is restricted by the intrinsic dependence of group velocity dispersion on fiber design. For example, achieving spectrally abrupt transitions - such as from deep normal dispersion at 1600 nm to deep anomalous dispersion at 1610 nm - is physically unfeasible, as the wavelength dispersion remains monotonic unless a resonance is introduced. Consequently, generating arbitrary dispersion landscapes is challenging, if not impossible, using programmable thermal tuning alone.

We emphasize, however, that the tuning range can be significantly extended by combining in-fiber dispersion control with input pulse shaping, such as tailoring the temporal intensity profile [link] or applying spectral phase control to femtosecond pulses. Modulating the input pulse amplitude and phase can alter the position of soliton fission along the fiber, shift the phase-

matching conditions for dispersive wave generation, and vary the achievable bandwidth for each pulse replica. When combined with local dispersion control, this approach can enable a broader range of seed events and four-wave mixing processes, potentially allowing on-demand spectral shaping across the entire bandwidth of a target supercontinuum.

To address the Reviewer's comment, we have extended the text in the Discussion section as follows: ... In addition, pulse shaping often requires complex and expensive equipment, whereas the proposed method provides a more accessible and straight-forward control framework. **Note that although local dispersion tuning is a powerful tool, it is fundamentally limited by physical constraints such as phase matching and soliton dynamics, as well as the inability to produce abrupt, steep GVD changes. This limitation can be overcome by combining input pulse shaping with local GVD control which may allow access to a broader range of seed events and four-wave mixing processes, potentially enabling on-demand spectral shaping across the full bandwidth of a target supercontinuum.** Mechanical deformation of multimode fibers provides an alternative approach that uses controlled bending to adjust dispersion characteristics and redistribute spectral power [4, 43]. ...

With that, I would suggest to accept this novel and inspiring work.

Thank you.

Response to comments of Reviewer #3

First, we would like to thank the Reviewer for thoroughly reading the revised manuscript and providing feedback on the revisions. Below, we address the comments raised by the Reviewer.

R3.1: Comment 1: The authors do not attempt to match simulation and experiments for a particular case as I suggested, but I concede that if they really don't know their incoupled power and their pulse shape, it may be a futile effort. I think it is important that these uncertainties are now clearly stated in the manuscript. As a minor point, I do not see why the presence of unheated sections could not have been included in simulations.

We thank the Reviewer for acknowledging that the uncertainties are now sufficiently addressed in the manuscript. We also appreciate the remark regarding the unmodulated fiber section. Including unheated fiber sections in the simulations would indeed be possible in principle by adjusting the modeled domain. However, as the primary aim of the simulations in this work is to illustrate the qualitative impact of temperature modulation with maximum flexibility in terms of available parameters, rather than to exactly replicate the experimental configuration, we chose a different model that assumes modulation along the entire fiber length. This approach allows to benchmark the capabilities of the system to shape output spectra without introducing constraints arising from the specific experimental setup used here.

To address this point more explicitly in the manuscript, we have revised the main text as follows: ... **Note that simulations are designed to highlight the system's capabilities rather than to precisely replicate experimental conditions. The simulations make unrestricted use of**

the parameter space, assuming precise temperature control, absence of thermal disturbances, and no initial unmodulated fiber section, thereby allowing maximum flexibility in all relevant parameters without introducing constraints from the specific setup used here. This strategy ensures that simulations and experiments complement each other effectively, while experimental optimization is conducted independently of the specific LCF configuration, enabling optimized output spectra even when the fiber parameters are not precisely known. ...

R3.2: Comment 2: I asked about the uncertainty of measured quantities, e.g. data in table 1, in light of the stated day-to-day variations. The authors do not answer this at all. They mostly discuss that the variations are not due to liquid degradation, which is an interesting point in its own right, but not really what I asked about. The authors argue that the uncertainties are not important because ultimately the setup could be fiber-integrated in a more robust way. I will leave it to the editor to consider whether this is an appropriate attitude to experimental data and measurements.

We thank the Reviewer for mentioning the point regarding the quantification of the uncertainty of the measured quantities in Tab. 1. To scientifically quantify the variations of these optimization parameters, a rigorous statistical analysis would be required. This would involve performing a large number of repeated measurements - likely exceeding 100 - to obtain statistically meaningful results. Given the current speed of the optimization, such an extensive dataset cannot be acquired within a reasonable timeframe. As noted in the manuscript, implementing experimental configurations that allow for faster optimization (e.g. via thermal printing heads, metallic wires, resistive microheaters, light-to-heat conversion) would make such an analysis feasible in the future.

To address the comment of the Reviewer, we have expanded the Discussion section of the manuscript as follows: ... In principle, this approach could be further accelerated by introducing light-absorbing materials (e.g. graphene) directly into the liquid core. **Note that reducing the optimization time would also allow quantification of the uncertainty of the measured optimization performance (Tab. 1 and 2) through a detailed statistical analysis, which is currently not feasible due to the comparably long duration of the optimization.** ...

R3.3: Comment 3: I think the authors have done a good job of answering this concern, which was also raised by other reviewers.

Thank you.

R3.4: Comment 4: I think the concern is adequately addressed, and that the information added to the manuscript is interesting and helpful.

Thank you.

R3.5: Comment 5: I think this point is adequately addressed by the additions to both the main text and the supplementary document. In fact, at closer inspection it appears to me that the carbon chlorides may be even better candidates for tunability, having similar thermo-optic coefficients, but a smaller refractive-index difference to silica.

We thank the Reviewer for the positive feedback on our response and for highlighting that carbon chlorides, for example in this study [link], may serve as alternative systems for implementing programmable fibers. We also appreciate the original comment from the first round of review, which encouraged us to explore a broader range of materials and material classes for this application. We are currently preparing a detailed simulation-based study to identify optimal material candidates for programmable fiber implementation, which we plan to submit soon. Thank you again for bringing this to our attention.

To address the comment of the Reviewer, the relevance of carbon chlorides has been incorporated into the new version of the manuscript as follows: ... Recent work has successfully demonstrated such integration in the context of stimulated Brillouin-Mandelstam scattering, highlighting their applicability in advanced fiber-based photonic systems [60]. **Note that carbon chlorides [link] may represent alternative candidates for implementing programmable fibers, as they exhibit high thermo-optic coefficients and a smaller refractive-index contrast with silica.** Another relevant type of waveguide are hybrid fibers containing unconventional materials [61], ...